# mGRPO: Unlocking LLM Reasoning through Multilingual Thinking

## Abstract

As LLMs develop stronger multilingual capabilities, the long-standing English-centric bias is gradually diminishing. In some reasoning tasks, responses in non-English languages even surpass those in English. Existing approaches, such as majority voting or weighting across languages, have explored this potential but often fall short due to task complexity and suboptimal language selection. To investigate the role of language diversity in reasoning, we conduct a *Polyglot Thinking Experiment*, prompting models to answer each question in ten different languages or without any language restriction. Results show that non-English responses often achieve higher accuracy than English ones, and the best performance frequently emerges when the model is free to choose its response language. These findings suggest that LLMs benefit from a broader and more flexible multilingual thinking space. Building on this insight, we propose **Multilingual Group Relative Policy Optimization (mGRPO)**, a reinforcement learning framework that improves LLM reasoning by generating multilingual preference data online using both language-constrained and unconstrained prompts. The model is optimized through group-wise reward comparisons based on accuracy and reasoning format. Despite relying on only ~18.1k training examples without chain-of-thought supervision, mGRPO achieves consistent gains across four benchmarks: MGSM, MATH500, PolyMath, and X-CSQA, outperforming two base LLMs (Qwen2.5 and Llama3) by an average of 7.5% and obtains SOTA performance. These results highlight the value of multilingual thinking and demonstrate that mGRPO provides a lightweight yet effective approach to unlock reasoning potential in LLMs.

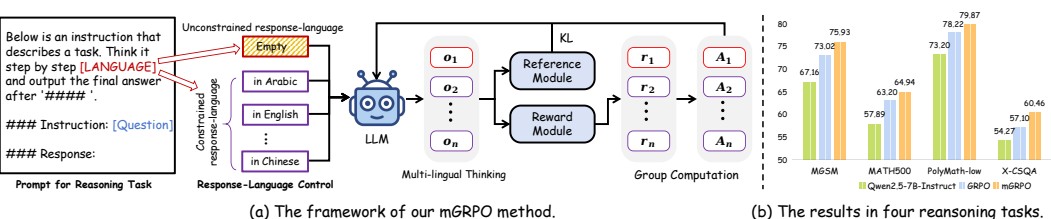

(a) The framework of our mGRPO method.   (b) The results in four reansoning tasks.

Figure 1: An Overview and Key Results. (a) Our mGRPO method's framework. It enhances the reasoning process by compelling the model to engage in **Multi-lingual Thinking**, which is then optimized via group relative policy. (b) A comparison of performance on four reasoning tasks. The results indicate that mGRPO achieves a significant improvement over both the Qwen2.5-7B-Instruct baseline and the standard GRPO approach (only output in English), demonstrating its efficacy.

## 1 Introduction

Large Language Models (LLMs) excel at a wide range of tasks, particularly reasoning (Jaech et al., 2024; DeepSeek-AI et al., 2025). However, they often display an English bias—achieving stronger performance with English inputs or responses (Chen et al., 2024; Shi et al., 2023; Huang et al., 2023; 2022). Recent advances suggest that this bias is weakening. LLMs trained on more diverse corpora increasingly demonstrate strong, and in some cases superior, reasoning abilities when operating in

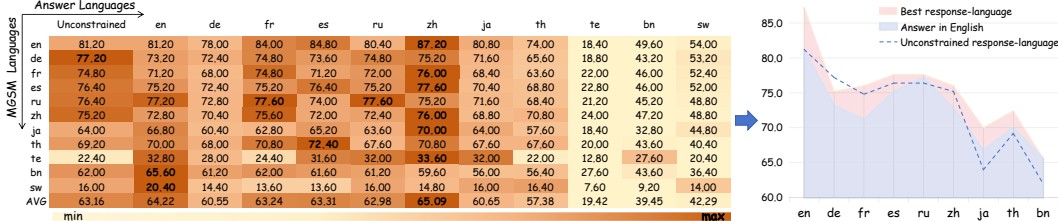

| MGSM Languages \ Answer Languages | Unconstrained | en | de | fr | es | ru | zh | ja | th | te | bn | sw |
|---|---|---|---|---|---|---|---|---|---|---|---|---|
| en | 81.20 | 81.20 | 78.00 | 84.00 | 84.80 | 80.40 | 87.20 | 80.80 | 74.00 | 18.40 | 49.60 | 54.00 |
| de | 77.20 | 73.20 | 72.40 | 74.80 | 73.60 | 74.80 | 75.20 | 71.60 | 65.60 | 18.80 | 43.20 | 53.20 |
| fr | 74.80 | 71.20 | 68.00 | 74.80 | 71.20 | 72.00 | 76.00 | 68.40 | 63.60 | 22.00 | 46.00 | 52.40 |
| es | 76.40 | 75.20 | 72.40 | 75.20 | 76.40 | 75.20 | 77.60 | 70.40 | 68.80 | 22.80 | 46.00 | 52.00 |
| ru | 76.40 | 77.20 | 77.60 | 74.00 | 77.60 | 75.20 | 77.60 | 71.60 | 68.40 | 21.20 | 45.20 | 48.80 |
| zh | 75.20 | 72.80 | 70.40 | 75.60 | 72.00 | 72.40 | 76.00 | 68.80 | 70.80 | 24.00 | 47.20 | 48.80 |
| ja | 64.00 | 66.80 | 60.40 | 62.80 | 65.20 | 63.60 | 70.00 | 64.00 | 57.60 | 18.40 | 32.80 | 44.80 |
| th | 69.20 | 70.00 | 68.00 | 70.80 | 72.40 | 67.60 | 70.80 | 67.60 | 67.60 | 20.00 | 43.60 | 40.40 |
| te | 22.40 | 32.80 | 28.00 | 24.40 | 31.60 | 32.00 | 33.60 | 32.00 | 22.00 | 12.80 | 27.60 | 20.40 |
| bn | 62.00 | 65.60 | 61.20 | 62.00 | 61.60 | 61.20 | 59.60 | 56.00 | 56.40 | 27.60 | 43.60 | 36.40 |
| sw | 16.00 | 20.40 | 14.40 | 13.60 | 13.60 | 16.00 | 14.80 | 16.00 | 16.40 | 7.60 | 9.20 | 14.00 |
| AVG | 63.16 | 64.22 | 60.55 | 63.24 | 63.31 | 62.98 | 65.09 | 60.65 | 57.38 | 19.42 | 39.45 | 42.29 |

Figure 2: *Polyglot Thinking Experiment* results (left part) on MGSM (Shi et al., 2023) in Qwen2.5-7B-Instruct (Yang et al., 2024a) model, including ten languages: English (en), German (de), French (fr), Spanish (es), Russian (ru), Chinese (zh), Japanese (Ja), Thai (th), Telugu (te), Bengali (Bn), and Swahili (sw). The right panel highlights the best score (red area) under specified-language settings, the score when responding in English (blue area), and the score when the response language is unconstrained (blue dashed line).

non-English languages (Qin et al., 2024; Zhu et al., 2024a; Dubey et al., 2024; Yang et al., 2024a; Aryabumi et al., 2024; Gao et al., 2025; Huang et al., 2025; Etxaniz et al., 2024).

Recent work suggests that multilingual thinking—the ability to reason across diverse languages—can enhance performance on complex tasks (Gao et al., 2025). Training-free approaches, such as majority voting (Qin et al., 2023) or automatic language selection with response weighting (Zhang et al., 2024), attempt to exploit this language diversity without model fine-tuning. Yet their effectiveness is often limited by task complexity and suboptimal language choices (Gao et al., 2025). Meanwhile, reinforcement learning (RL)-based methods, such as PPO (Schulman et al., 2017), or RL-like approaches such as DPO (Rafailov et al., 2023) and GRPO (DeepSeek-AI et al., 2025), have shown promise in improving LLM reasoning. However, most existing RL training relies on English-centric or English-estimated preference data (Yang et al., 2024c; She et al., 2024), which restricts models from fully benefiting from multilingual thinking, particularly when English is not the most effective reasoning language.

Previous observations suggest that for certain tasks, reasoning in non-English languages can outperform English. To systematically investigate this effect, we introduce a *Polyglot Thinking Experiment* on MGSM (Shi et al., 2023). In this setup, for each language in MGSM, we construct prompts that elicit responses in both unconstrained response-languages, where the model can freely choose the output language, and constrained response-languages settings. The detailed prompt design is illustrated in Figure 1(a), and the corresponding results are shown in Figure 2. Our findings indicate that, in constrained response-language settings, Chinese responses, on average, outperform English, while no single setting consistently dominates across all languages. Notably, under the unconstrained response-language setting, models often outperform the English-only baseline. We observe that this is enabled by a flexible reasoning space, manifested in the use of multiple response languages and code-switching—where responses mix surface entities (e.g., names of people or places) from question with English or Chinese. These results suggest that allowing the model to operate without strict language constraints expands its thinking space and flexibility.

Motivated by this insight, we combine language-constrained and unconstrained prompts to form preference groups that capture diverse multilingual thinking variations. Building on GRPO (DeepSeek-AI et al., 2025), we propose multilingual GRPO (mGRPO), a reinforcement learning method that explicitly leverages this multilingual thinking space to enhance LLM reasoning. As shown in Figure 3, mGRPO consists of three components: the Polyglot Thinking Generation Module, the Reward Module, and the Group Relative Policy Optimization Module. For each question, we generate a group of responses—one under an unconstrained setting and others in randomly assigned target languages—thereby creating diverse multilingual thinking data. The reward function is rule-based, combining correctness (measured by the final answer) and format (encouraging reasoning steps). Based on these reward scores, group-relative advantages are computed to establish preference rankings, which guide policy optimization through GRPO.

We evaluated mGRPO on four reasoning benchmarks: MGSM (Shi et al., 2023), mMATH (Lightman et al., 2023), PolyMath (Wang et al., 2025), and X-CSQA (Lin et al., 2021), covering 23 languages.

Using ∼18k multilingual mathematics training examples and training on the Qwen2.5-7B-Instruct model, mGRPO achieves average improvements of 2.91%, 1.74%, 1.65%, and 3.36% over the standard GRPO approach (which only outputs in English) on the four benchmarks, respectively, as shown in Figure 1(b).

Our contributions are summarized as follows:

- We reveal that English is not always the best response language in reasoning tasks, and that unconstrained language responses often yield surprising results. Based on this, we propose leveraging multilingual thinking to enhance LLM reasoning capabilities.

- We introduce **mGRPO**, a novel reinforcement learning framework that online generates multilingual preference sets constructed from multilingual thinking to optimize LLMs.

- Through experiments on four reasoning benchmarks and two base models, mGRPO significantly improves LLM performance on both mathematical and commonsense reasoning tasks, demonstrating the powerful impact of multilingual thinking in enhancing LLM reasoning abilities.

## 2 RELATED WORK

**Multilingual Thinking of LLMs.** Early LLMs were predominantly trained on English-centric data, resulting in better performance when questions or responses were in English (Shi et al., 2023). To improve reasoning capabilities in other languages, recent work has proposed cross-lingual chain-of-thought (CoT) prompting strategies (Ranaldi & Zanzotto, 2023; Huang et al., 2023). From a training perspective, beyond merely increasing multilingual training data, some studies translate English questions (Huang et al., 2024; Zhu et al., 2024b) or CoT responses (Chen et al., 2024; Lai & Nissim, 2024; Chai et al., 2025) into multiple languages and fine-tune models on the augmented data. Besides translation-based methods, multilingual preference training has gained traction (She et al., 2024; Yang et al., 2024c), often treating English reasoning as the reference to guid multilingual outputs. However, as multilingual LLMs improve, their reasoning in certain languages can surpass English (Gao et al., 2025), sparking growing interest in leveraging multilingual thinking to boost overall performance.

**Enhancing LLM Reasoning with Multilingual Thinking.** Gao et al. (2025) showed that aggregating reasoning across k languages (Acc@k) can outperform English-only reasoning by up to 10 points, with robustness to both translation quality and language selection. Building on similar insights, Qin et al. (2023) proposed cross-lingual prompting, which first guides the model to understand the question in English before generating answers in multiple languages, with final predictions determined by majority voting. However, their method suffers from instability due to arbitrary language choices. To address this, AutoCAP (Zhang et al., 2024) introduces an automated scheme in which the LLM selects languages and assigns weights to CoTs, generating final answers through weighted multilingual outputs. Despite these efforts, such approaches remain limited by task complexity and generalization challenges.

In contrast, our method, mGRPO, adopts a reinforcement learning framework that allows the model to internally explore and integrate multilingual thinking without relying on post-hoc voting or language-specific heuristics, while improving generalization with minimal supervision (i.e., only gold answers). It supports online generation of preference data during training, scales effectively across model sizes, and achieves consistent gains in both high- and low-resource language settings.

## 3 METHOD

Motivated by the diverse performance exhibited in multilingual thinking, we aim to let the model learn from such diversity instead of aligning all reasoning to English. We propose a multilingual reinforcement learning framework, mGRPO (Multilingual Group Relative Policy Optimization), to enhance LLMs' reasoning abilities through multilingual thinking. As illustrated in Figure 3, mGRPO consists of three modules: (1) Polyglot Reasoning Generation Module (§ 3.1), (2) Reward Module (§ 3.2), and (3) Group Relative Policy Optimization Module (§ 3.3).

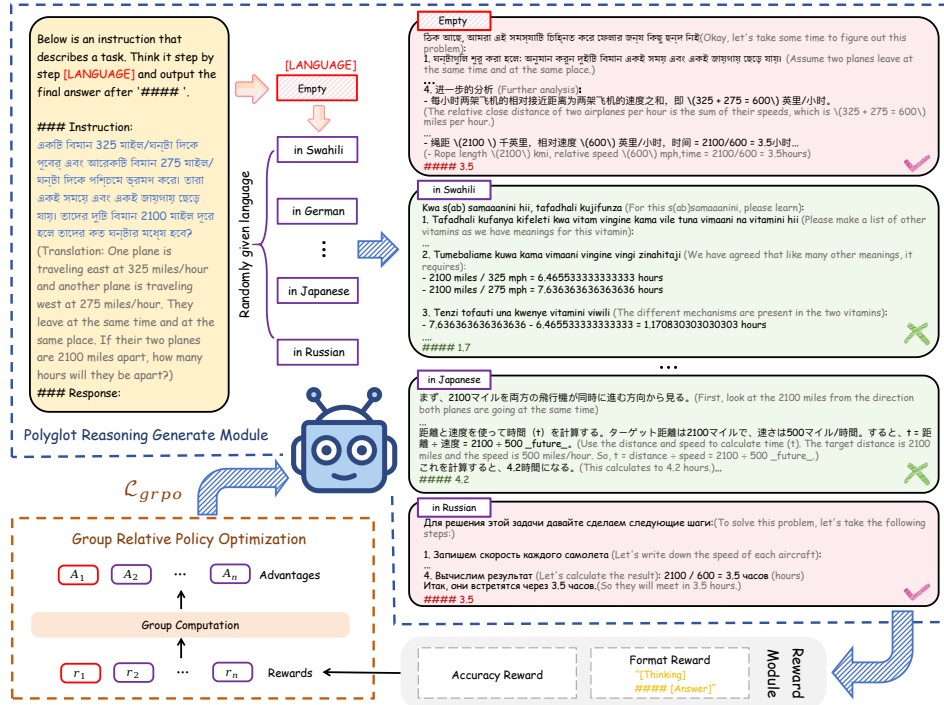

Figure 3: The framework of mGRPO, including Polyglot Reasoning Generate Module (PRGM), Reward Module and Group Relative Policy Optimization Module.

## 3.1 POLYGLOT REASONING GENERATION MODULE

GRPO (DeepSeek-AI et al., 2025) is a reinforcement learning method that improves upon PPO by removing the value function and estimating advantages in a group-relative manner. To construct the training group for a question-answer pair $(q, a) \in \mathcal{D}$, it sample $n$ responses $\{o_i\}_{i=1}^n$ from the old policy $\pi_{\theta_{\text{ref}}}$.

Our proposed **Polyglot Reasoning Generation Module (PRGM)** is designed to guide the LLM in generating a group of $n$ multilingual responses for each input. As shown in the upper part of Figure 3, given an input question, we generate a set of $n$ responses using prompts $\{p_i\}_{i=1}^n$ with or without explicit language instructions. Specifically, one response is generated with no language constraint (i.e. "[LANGUAGE]" is empty), while the remaining responses are generated using prompts that specify a reasoning language randomly chosen from a predefined languages set. These responses form the *Multilingual Thinking* set $\{o_i\}_{i=1}^n$, which may include both correct and incorrect answers.

This module operates **online during training**, enabling the model to continuously generate fresh multilingual responses. Such an online approach reduces storage overhead than (She et al., 2024; Yang et al., 2024c) and facilitates broader exploration of the multilingual thinking space.

## 3.2 REWARD MODULE

Each response $o_i$ is then evaluated using a reward module to obtain an individual reward $r_i$. To assess response quality, we design a rule-based reward function composed of two parts $r_i = \text{AR}(o_i) + \text{FR}(o_i)$, where:

- **Accuracy Reward (AR):** A binary reward that evaluates whether the final predicted answer exactly matches the gold-standard answer. Formally,

$$\text{AR}(o_i) = \begin{cases} 1, & \text{if Answer}(o_i) = \text{Gold}(q) \\ 0, & \text{otherwise} \end{cases} \tag{1}$$

- **Format Reward (FR):** A binary reward that encourages structured reasoning. It returns 1 if the response contains a reasoning process (denoted by the keyword `[Thinking]`) and presents the final answer in the required format (i.e., following `"#### "`). Formally,

$$\text{FR}(o_i) = \begin{cases} 1, & \text{if } o_i \text{ contains } \texttt{[Thinking]} \text{ and } \texttt{[Answer]} \text{ follows } \texttt{"#### "} \\ 0, & \text{otherwise} \end{cases} \tag{2}$$

To prevent the model from taking shortcuts, e.g., generating minimal text before directly outputting the answer, we additionally enforce a minimum length constraint of 100 characters for the reasoning content within the `[Thinking]` section as part of the format reward.

The final reward $r_i \in \{0, 1, 2\}$ thus encourages both correctness and structured reasoning format, without requiring annotated reasoning steps.

### 3.3 GROUP RELATIVE POLICY OPTIMIZATION MODULE

Then, we follows the GRPO (DeepSeek-AI et al., 2025) to optimization our model as shown in left-downer corner of Figure 3 and right part of Figure 1(a). The advantage $A_i$ of the $i$-th response is calculated by normalizing the rewards $\{r_i\}_{i=1}^n$ of the group:

$$A_i = \frac{r_i - \text{mean}(\{r_i\}_{i=1}^n)}{\text{std}(\{r_i\}_{i=1}^n)}. \tag{3}$$

GRPO adopts a PPO-style clipped objective, with a KL penalty between the current policy $\pi_\theta$ and the reference model $\pi_{\theta_{\text{ref}}}$ directly integrated into the loss to simplify training.

So, the loss of our mGRPO is:

$$\mathcal{L}_{\text{mGRPO}}(\theta) = \mathbb{E}_{(q,a)\sim\mathcal{D},\{o_i\}_{i=1}^n\sim\pi_{\theta_{\text{ref}}}(o_i|p_i,q)_{i=1}^n} \Bigg[$$

$$\frac{1}{n}\sum_{i=1}^{n}\frac{1}{|o_i|}\sum_{t=1}^{|o_i|}\left\{\min\left[\frac{\pi_\theta^{i,t}}{\pi_{\theta_{\text{ref}}}^{i,t}}\hat{A}_i, \text{clip}\left(\frac{\pi_\theta^{i,t}}{\pi_{\theta_{\text{ref}}}^{i,t}}, 1-\epsilon, 1+\epsilon\right)\hat{A}_i\right] - \beta\mathbb{D}_{\text{KL}}(\pi_\theta\|\pi_{\text{ref}})\right\}\Bigg] \tag{4}$$

where $\pi^{i,t}$ denotes the conditional probability of the token at position $t$, formally:

$$\pi^{i,t} = \pi(o_{i,t}|p_i, q, o_{i,<t}), \tag{5}$$

where $p_i$ is the $i$-th prompt with or without explicit language instructions to obtain $o_i$.

Compared with previous approaches that rely on supervised translations (She et al., 2024) or fixed language anchors (Yang et al., 2024c), mGRPO enables LLMs to autonomously explore and learn from multilingual thinking behaviors, promoting a more flexible and effective reasoning paradigm.

## 4 EXPERIMENTS

### 4.1 DATASETS

**Training Datasets.** We use the mathematical reasoning dataset from MAPO (She et al., 2024) as training data. It consists of 1,703 English questions from a subset of NumGLUE (Mishra et al., 2022), together with ChatGPT-translated versions in nine languages, including Bengali (BN), Thai (TH), Swahili (SW), Japanese (JA), Chinese (ZH), Russian (RU), German (DE), Spanish (ES), and French (FR), resulting in a total of 18,140 examples.

**Benchmarks.** Our evaluation is based on three mathematical reasoning benchmarks (MGSM, MATH500, and PolyMath) and one commonsense reasoning benchmark (X-CSQA) to assess improvements in LLM reasoning abilities. **MGSM** (Shi et al., 2023) serves as an in-domain benchmark, derived from 250 GSM8K (Chen et al., 2024) test samples translated by native speakers into 10 typologically diverse languages. **MATH500** (Lightman et al., 2023) is an out-of-domain benchmark consisting of 500 diverse mathematical problems in English, with six additional translated versions

Table 1: The results in 4 multilingual reasoning benchmarks. Languages in MGSM are categorized into high-resource (HRL: ZH, FR, DE, JA, RU, ES) and underrepresented-resource (URL: BN, SW, TE (Telugu), TH) groups based on their presence in pretraining corpora such as mC4 (Xue et al., 2021). AVG represents the average performance of all languages in a benchmark. Best score in **bold**.

| Model | MGSM | | | MATH500 | | PolyMath | | | | X-CSQA |
|---|---|---|---|---|---|---|---|---|---|---|
| | AVG | HRL | URL | EN | AVG | low | medium | high | top | AVG |
| *Qwen2.5-7B-Instruct* | | | | | | | | | | |
| Base (Yang et al., 2024a) | 67.16 | 75.73 | 48.80 | 70.80 | 57.89 | 73.20 | 23.69 | 9.02 | 5.07 | 54.27 |
| xRFT (Chen et al., 2024) | 68.47 | 81.07 | 43.20 | 73.40 | 53.12 | 60.80 | 18.40 | 7.69 | **7.73** | 49.25 |
| LIDR (Yang et al., 2024c) | 69.60 | 79.07 | 50.30 | 73.20 | 62.46 | 74.93 | 25.07 | **9.87** | 4.62 | 53.18 |
| MAPO (She et al., 2024) | 66.29 | 75.80 | 47.40 | 76.20 | 61.20 | 76.31 | 23.24 | 8.04 | 5.87 | 50.69 |
| GRPO (DeepSeek-AI et al., 2025) | 73.02 | 81.07 | 56.60 | 74.80 | 63.20 | 78.22 | 23.42 | 8.84 | 6.36 | 57.10 |
| mGRPO (Ours) | **75.93** | **84.40** | **58.70** | **76.80** | **64.94** | **79.87** | **25.24** | **9.87** | 7.07 | **60.46** |
| *Llama3-8B-Instruct* | | | | | | | | | | |
| Base (Dubey et al., 2024) | 52.22 | 57.33 | 37.70 | 29.20 | 26.00 | 60.44 | 4.84 | 1.91 | 2.53 | 45.12 |
| xRFT (Chen et al., 2024; She et al., 2024) | 53.89 | 58.40 | 42.30 | 27.40 | 23.37 | 50.62 | 4.93 | 2.44 | 1.96 | 48.15 |
| LIDR (Yang et al., 2024c) | 55.53 | 58.47 | 45.10 | 28.00 | 24.31 | 52.71 | 5.02 | 2.00 | 2.18 | 52.22 |
| MAPO (She et al., 2024) | 60.69 | 63.93 | 50.90 | 30.40 | 25.26 | 58.84 | 4.31 | 1.73 | 2.76 | 43.89 |
| GRPO (DeepSeek-AI et al., 2025) | 64.58 | 68.80 | 54.20 | 30.00 | 24.43 | 53.87 | 4.76 | 2.44 | **3.82** | 53.36 |
| mGRPO (Ours) | **68.11** | **72.33** | **58.30** | **32.00** | **26.71** | **66.48** | **5.51** | **2.76** | 3.42 | **53.64** |

included for multilingual evaluation. **PolyMath** (Wang et al., 2025) provides a multilingual reasoning benchmark with 9,000 math problems across 18 languages with four difficulty levels. **X-CSQA** (Lin et al., 2021) extends CSQA to 16 languages and challenges models to interpret complex logical relations expressed across diverse linguistic forms. The total number of evaluation languages is 23, and the details of the languages covered by each benchmark can be found in Appendix B.

## 4.2 EXPERIMENTAL SETUP

**Base Models and Baselines.** We evaluate mGRPO on the Qwen2.5-7B-Instruct(Yang et al., 2024a) and Llama3-8B-Instruct (Dubey et al., 2024) models. For baselines, we compare mGRPO with several strong methods: (1) **xRFT** (Yuan et al., 2023), a rejection sampling-based method using CoT traces generated and translated from Qwen-Math-7B-Instruct (Yang et al., 2024b); (2) **MAPO** (She et al., 2024), which aligns multilingual thinking paths to English through translation-estimated-based preference optimization; (3) **LIDR** (Yang et al., 2024c), which employs self-improving DPO training based on performance disparities between non-English languages and English; and (4) **GRPO** (DeepSeek-AI et al., 2025), which uses our multilingual training data and only generates English responses, to compare the influence of multilingual thinking on LLM reasoning. Full training configurations and data construction details are provided in Appendix C.

**Training Details.** For PRGM, we use a 10-language set to guide the roll-out, aligned with the languages in the training data. The roll-out is set to $n = 5$, including one non-language-constrained response and four responses in randomly selected languages from the set. Training is implemented using the verl[1] RL framework. For Qwen2.5-7B-Instruct, mGRPO is trained for 5 epochs with a learning rate of $1e-6$ and a batch size of 256. For the Llama3-8B-Instruct base model, mGRPO is trained for 1 epoch with the same settings. All models are trained using 8 NVIDIA A100 GPUs.

**Inference Setup.** At inference time, we use the same prompt format as during training (as shown in the left part of Figure 1(a)), leaving the language token [LANGUAGE] empty to allow the model to freely choose its response language. Reasoning steps are generated via greedy decoding. Final answers are extracted using rule-based parsing and evaluated using accuracy as the main metric.

## 4.3 RESULTS

We systematically evaluated mGRPO's performance on two mainstream baseline models. Table 1 presents the results on four reasoning benchmarks. Our method outperforms existing approaches in both reasoning performance and generalization across different difficulty levels. Per-language results for all test sets are provided in Appendix D.

---

[1]https://github.com/volcengine/verl

Based on Qwen2.5-7B-Instruct, mGRPO achieved state-of-the-art or competitive results on all tasks. Specifically, it achieved an average accuracy of 75.93% on the MGSM dataset, an 8.76% improvement over the base model, and outperformed GRPO by 2.1% on low-resource languages (i.e., URL). On MATH500 and its six language translation versions, mGRPO achieved an average score of 64.94%, surpassing GRPO and LIDR by 1.74% and 2.48%, respectively.

Table 2 reports that on the 134 most difficult examples in MATH500, mGRPO outperformed the strongest LIDR baseline by 2.9%. This advantage is primarily evident in higher-resource languages. For example, on Japanese and Turkish, mGRPO surpasses the LIDR method by 5.2% and 6.0%, respectively. Furthermore, across all languages, mGRPO surpasses the original GRPO setup, demonstrating that learning multilingual thinking has indeed stimulated stronger reasoning capabilities. On the latest PolyMath, due to the small size of the model, all methods failed to achieve significant improvements on tasks above medium. Therefore, we focused on the "low" difficulty tasks. Table 3 reports the performance of mGRPO on 18 languages in PolyMath-low, achieving state-of-the-art performance on 14 of them. Furthermore, on the commonsense reasoning task X-CSQA, mGRPO achieves a 3.36% improvement over the strongest baseline GRPO, validating the effectiveness of multilingual thinking in improving more general reasoning capabilities.

Table 2: The results of MATH500 on Qwen2.5-7B-Instruct with the 134 hardest examples.

| Model | AVG | EN | IT | JA | TR | ZH | TE | SW |
|---|---|---|---|---|---|---|---|---|
| *Qwen2.5-7B-Instruct* | | | | | | | | |
| Base | 33.3 | 46.3 | 45.5 | 29.1 | 21.6 | 41.8 | 28.4 | 20.2 |
| xRFT | 30.6 | 50.8 | 38.8 | 34.3 | 28.4 | 36.6 | 14.9 | 10.5 |
| LIDR | 39.8 | 51.5 | 49.3 | 44.8 | 38.8 | 38.8 | **32.8** | **22.4** |
| MAPO | 37.0 | 54.5 | 43.3 | 32.1 | 30.6 | 46.3 | 30.6 | 21.6 |
| GRPO | 37.5 | 50.8 | 44.0 | 42.5 | 40.3 | 38.1 | 29.9 | 17.2 |
| mGRPO (Ours) | **42.9** | **53.7** | **50.8** | **50.0** | **44.8** | **47.8** | 32.1 | 20.9 |

Table 3: The results in PolyMath-low across 18 languages.

| Model | AVG | EN | ZH | ES | AR | FR | BN | PT | RU | ID | DE | JA | SW | VI | IT | TE | KO | TH | MS |
|---|---|---|---|---|---|---|---|---|---|---|---|---|---|---|---|---|---|---|---|
| *Qwen2.5-7B-Instruct* | | | | | | | | | | | | | | | | | | | |
| Base | 74.6 | 89.6 | 79.2 | 87.2 | 80.0 | 84.0 | 66.4 | 80.8 | 83.2 | 81.6 | 76.0 | 68.8 | 14.4 | 79.2 | 83.2 | 36.8 | 74.4 | 73.6 | 79.2 |
| xRFT | 63.1 | 83.2 | 72.8 | 70.4 | 66.4 | 68.0 | 50.4 | 68.0 | 68.8 | 62.4 | 65.6 | 64.8 | 10.4 | 68.8 | 68.0 | 20.0 | 67.2 | 54.4 | 64.8 |
| LIDR | 76.4 | 89.6 | **84.8** | 84.8 | 80.8 | **86.4** | 66.4 | 80.0 | 84.0 | 83.2 | 80.8 | 75.2 | 17.6 | 80.0 | 81.6 | 38.4 | 76.8 | 76.8 | 81.6 |
| MAPO | 77.7 | 92.0 | 81.6 | 88.0 | 80.8 | 84.8 | 67.2 | 80.0 | **89.6** | 84.8 | 77.6 | 75.2 | 23.2 | **85.6** | 85.6 | 43.2 | 77.6 | 74.4 | 82.4 |
| GRPO | 79.7 | 92.0 | 84.0 | 86.4 | **87.2** | 82.4 | 71.2 | 85.6 | 88.0 | **87.2** | 80.8 | 75.2 | 32.8 | 83.2 | 87.2 | **44.8** | **81.6** | 76.0 | 82.4 |
| mGRPO (Ours) | **81.5** | **94.4** | 83.2 | **88.8** | 87.2 | 86.4 | **76.0** | **88.0** | 87.2 | 85.6 | **82.4** | **78.4** | 36.0 | 85.6 | **88.8** | **44.8** | 80.0 | **80.8** | **84.0** |
| *Llama3-8B-Instruct* | | | | | | | | | | | | | | | | | | | |
| Base | 61.4 | 73.6 | 54.4 | 65.6 | 59.2 | **69.6** | 52.8 | 67.2 | 62.4 | 68.8 | 60.8 | 57.6 | 39.2 | **67.2** | 69.6 | 38.4 | 59.2 | 56.0 | **66.4** |
| xRFT | 51.3 | 65.6 | 42.4 | 56.0 | 53.6 | 52.0 | 40.0 | 54.4 | 62.4 | 53.6 | 53.6 | 45.6 | 36.0 | 51.2 | 63.2 | 34.4 | 48.8 | 52.0 | 46.4 |
| LIDR | 54.1 | 61.6 | 49.6 | 59.2 | 50.4 | 60.8 | 44.0 | 65.6 | 58.4 | 56.0 | 56.8 | 45.6 | 37.6 | 57.6 | 58.4 | 36.8 | 47.2 | 52.0 | 51.2 |
| MAPO | 59.7 | 72.8 | 58.4 | 68.8 | 57.6 | 56.8 | 52.0 | 69.6 | 61.6 | 62.4 | 64.8 | 52.0 | 39.2 | 60.0 | 67.2 | 41.6 | 59.2 | 56.0 | 56.0 |
| GRPO | 55.0 | 68.0 | 60.0 | 56.0 | 51.2 | 60.8 | 39.2 | 63.2 | 62.4 | 50.4 | 63.2 | 53.6 | 34.4 | 52.0 | 62.4 | 39.2 | 48.8 | 57.6 | 47.2 |
| mGRPO (Ours) | **67.6** | **78.4** | **71.2** | **72.0** | **65.6** | 68.8 | **57.6** | **74.2** | **70.4** | **69.6** | **72.8** | **61.6** | **52.0** | 64.8 | **74.4** | **48.8** | **64.8** | **63.2** | **66.4** |

On Llama3-8B-Instruct, despite the original model's limited multilingual capabilities, mGRPO demonstrates strong potential for improvement. It achieved nearly a 16% improvement over the base model on MGSM and maintained a significant advantage on the more challenging MATH500. On PolyMath-low, mGRPO was the only method to surpass the base model, achieving substantial performance gains across multiple languages. For example, mGRPO improved over the base model by 16.8%, 12.8%, and 10.4% on Chinese, Swahili, and Telugu, respectively. mGRPO also performed remarkably well on X-CSQA, similar to GRPO, improving the base model's performance by 8.52%.

Overall, mGRPO demonstrated comprehensive superiority on both LLMs with different architectures, highlighting the versatility and robustness of our approach. Compared to GRPO, our mGRPO method further improved multiple key metrics, fully demonstrating the effectiveness of multilingual thinking in enhancing model reasoning performance.

## 5 ANALYSIS

### 5.1 ABLATION STUDY

We conduct ablation studies on the Qwen2.5-7B-Instruct model and evaluate it on the MGSM benchmark. First, we examine the impact of the format reward (i.e., w/o format reward). Next, we compare three PRGM roll-out variants: (1) without the unconstrained response-language roll-out (i.e., w/o unconstrained response); (2) with only the unconstrained response-language setting for

roll-out (i.e., only roll-out unconstrained response); and (3) only the English response roll-out (i.e., GRPO setting). Additionally, to assess the performance of our method on smaller models, we include two other sizes of Qwen2.5-Instruct models, with 1.5B and 3B parameters, respectively. Results are shown in Figure 4, with further details in Appendix E.

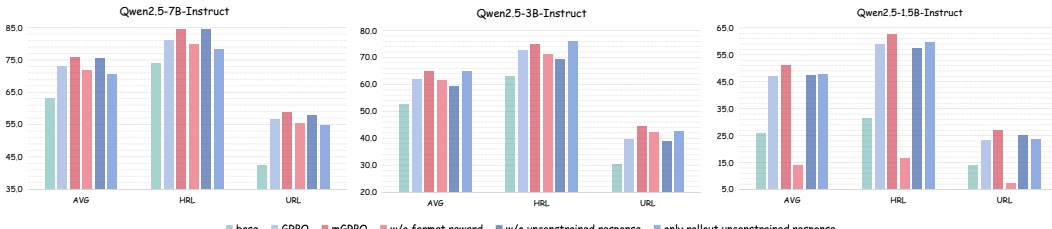

Figure 4: Ablation results on MGSM with three sizes of Qwen2.5-Instruct models.

The ablation studies validate the importance of each setting for the effectiveness of mGRPO. Based on the results and model behavior, we can make three observations:

**Format reward is crucial for guiding LLMs to generate multilingual thinking paths and valid final answers.** Without the format reward, during the roll-out phase of PRGM, the model often degenerates into uncontrolled behavior in low-resource languages (e.g., Thai, Swahili), such as skipping reasoning steps, adding irrelevant text, or reproducing the same content.

**Response without constrained language benefits smaller LLMs, while language-specified responses help improve performance in low-resource languages.** We observe that when all roll-outs are language-constrained, the performance of the 1.5B and 3B models drops significantly. The unconstrained language roll-out improves high-resource languages (e.g., a 1.1% gain on the 3B model), but causes a 1.6% drop in low-resource settings. Both response-language settings are important for constructing the multilingual thinking roll-out.

**Multilingual thinking unlocks more powerful LLM reasoning capabilities.** We find that later training roll-outs of mGRPO often converge to English. To verify if improvements come only from English reasoning, we ran experiments restricting all responses to English (i.e., the original GRPO setting). Its performance consistently falls short of our multilingual thinking roll-out setting, especially for smaller models.

## 5.2 LANGUAGE SET IN PRGM

The "languages" of multilingual thinking in PRGM are primarily randomly selected from a language set. To align with the MAPO and LIDR methods, our language set consists of all 10 languages in the training data, including both high-resource languages and low-resource languages (as defined in MGSM). Observing the impact of different language sets on mGRPO can also help us better understand the robustness or bias of our method with respect to language selection.

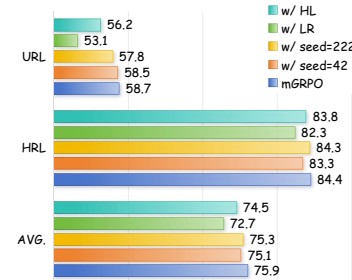

Figure 5: The results in MGSM with different language set for multilingual thinking base on Qwen2.5-7B-Instruct.

First, we established two language sets, low-resource (LR) and high-resource (HR) languages, each consisting of 10 languages and determined by their presence in pretraining corpora (such as mC4 (Xue et al., 2021) mentioned in MGSM). The LR set includes Bengali (BN), Thai (TH), Swahili (SW), Telugu (TE), Vietnamese (VI), Basque (EU), Arabic (AR), Hindi (HI), Urdu (UR), and Turkish (TR). The HR set includes Italian (IT), Chinese (ZH), English (EN), French (FR), German (DE), Japanese (JA), Russian (RU), Spanish (ES), Korean (KO), and Portuguese (PT). Models trained with HR or LR languages are

referred to as mGRPO w/ HR and mGRPO w/ LR, respectively. Second, to test the robustness of mGRPO with a mix of HR and LR languages, we created two additional sets (random seeds 42 and 222), each randomly sampling 5 HR and 5 LR languages: (1)Seed=42: [EN, KO, ZH, DE, FR, VI, BN, TR, TH, AR]; (2)Seed=222: [FR, ES, IT, KO, JA, SW, TE, VI, BN, UR].

The results on MGSM are shown in Figure 5. We observed that using only HR languages for multilingual thinking obtained comparable performance compared to the original setting. However, using only LR languages limited the performance gains. When the 10-language set included both HR and LR languages, the differences caused by language selection were reduced. The set with significant differences (seed=222) performed slightly worse overall due to the omission of the primary language EN and the resource-rich language ZH. We further experiment on the impact of different language quantities and roll-out values on performance, with results reported in the Appendix F and G, respectively.

### 5.3 How Many Language Are Utilized During the Reasoning Process

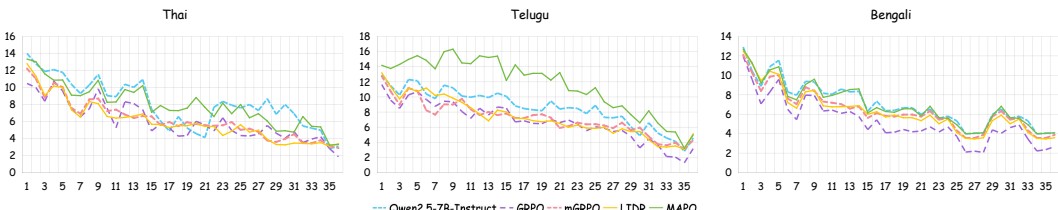

Figure 6: Layer-wise statistics of the number of distinct languages present in each tokens.

Since mGRPO tends to converge toward English reasoning in the later stages of training (detailed analysis can be seen in Appendix H)—and predominantly generates English CoT during inference—we hypothesize that the model integrates multilingual thinking paths into a unified English-centric latent space. Consequently, we expect it to rely on fewer non-English language tokens during reasoning.

To verify this, we adopt a logit lens–based analysis (Wang, 2025) to examine the token activations at the decoding step on each layer. After excluding generic digits and punctuation, we use the `langid`[2] toolkit to identify the language of each token and compute the number of distinct languages used per layer. This analysis is performed on the mGRPO model based on Qwen2.5-7B-Instruct, evaluated on the first 50 MGSM examples in three low-resource languages. The average results, shown in Figure6, indicate that mGRPO consistently activates fewer language types compared to baselines—supporting our hypothesis that **multilingual thinking has been fused into an English-dominant latent space that facilitates stronger reasoning capabilities in our method.** So the reasoning path is almost generated in English. We explore a simple test-time strategy (see Appendix I) that enables mGRPO to achieve a significantly higher language accuracy with only a minor performance trade-off.

## 6 Conclusion

This work introduces mGRPO, a reinforcement learning framework that enhances reasoning in LLMs by leveraging multilingual thinking. By generating polyglot reasoning paths and optimizing accuracy- and format-aware rewards, mGRPO encourages models to internalize multilingual thinking strategies. Our results demonstrate that mGRPO improves performance on four reasoning tasks across 23 languages using both Qwen2.5 and Llama3 architectures. It achieves an average 7.5% improvement over two base LLMs on MGSM, multilingual-version MATH500, and PolyMath-low, while preserving generalization to non-mathematical domains. Analysis shows that the model gradually shifts from multilingual to English reasoning during training, achieving better performance than training solely in English. This suggests that multilingual thinking traces act as scaffolding for stronger, language-agnostic reasoning capabilities. However, the model still tends to favor English during reasoning, prompting us to introduce a simple test-time strategy to balance performance gains with improved language consistency. This is a direction worth exploring in future research.

---

[2]https://github.com/saffsd/langid.py

## ETHICS STATEMENT

This work does not involve human subjects, personal data, or sensitive information. All datasets used in our experiments (training data from MAPO, test data from MGSM, MATH500, and its open-source translations, PolyMath, and X-CSQA) are publicly available and intended solely to enhance and evaluate LLM reasoning capabilities. We strictly adhere to ethical research practices and did not perform any data collection that could raise privacy, safety, or fairness concerns. Our approach improves reasoning by leveraging multilingual thinking generated by the models themselves, without introducing risks of harmful applications. To the best of our knowledge, this research complies with the ICLR ethical guidelines and presents no foreseeable ethical issues.

## REPRODUCIBILITY STATEMENT

We have made substantial efforts to ensure the reproducibility of our work. Detailed dataset descriptions can be found in Section 4.1 and Appendix B, while training configurations and hyper-parameters are reported in Section 4.2 and Appendix C. As our method is implemented on the open-source VERL framework, it can be clearly reproduced through our settings for multilingual thinking outputs and reward functions from Section 3.1 and 3.2. Upon acceptance of this paper, we will release our models along with the training and inference code to facilitate replication and further research.

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

## A  LLM USAGE

In this section, we clarify the role of LLMs in this work. The model was used solely for language polishing, including improving grammar, style, and readability, and did not contribute to the research design, analysis, or conclusions.

## B  BENCHMARKS AND COVERED LANGUAGES

In this section, we provide the languages covered by the benchmarks used in our evaluation, as shown in Table 4.

Table 4: Summary of benchmarks and their covered languages.

| Dataset | #Languages | Languages |
|---|---|---|
| MGSM (Shi et al., 2023) | 10 | Chinese (ZH), French (FR), German (DE), Japanese (JA), Russian (RU), Spanish (ES) (HRL); Bengali (BN), Swahili (SW), Telugu (TE), Thai (TH) (URL) |
| MATH500 (Lightman et al., 2023) | 7 | English (EN), Chinese (ZH), Japanese (JA), Telugu (TE), Swahili (SW), Italian (IT), Turkish (TR) |
| PolyMath (Wang et al., 2025) | 18 | English (EN), Chinese (ZH), Spanish (ES), Arabic (AR), French (FR), Bengali (BN), Portuguese (PT), Russian (RU), Indonesian (ID), German (DE), Japanese (JA), Swahili (SW), Vietnamese (VI), Italian (IT), Telugu (TE), Korean (KO), Thai (TH), Malay (MS) |
| X-CSQA (Lin et al., 2021) | 16 | Arabic (AR), German (DE), English (EN), Spanish (ES), French (FR), Hindi (HI), Italian (IT), Japanese (JA), Dutch (NL), Polish (PL), Portuguese (PT), Russian (RU), Swahili (SW), Urdu (UR), Vietnamese (VI), Chinese (ZH) |

For MATH500 benchmark, the ZH, JA, TE, and SW versions of MATH500 are from https://huggingface.co/datasets/appier-ai-research; the IT and TR versions are from https://huggingface.co/datasets/bezir/MATH-500-multilingual.

## C   BASELINES

We compare mGRPO with several strong baselines:

- The **base model**, Qwen2.5-7B-Instruct and Llama3-8B-Instruct, already demonstrates strong performance on reasoning tasks, which serves as a solid reference point.

- **xRFT** (Yuan et al., 2023) is a rejection sampling–based fine-tuning approach. It uses CoT traces generated by Qwen2.5-Math-7B-Instruct (Yang et al., 2024b), translated into multiple languages. After filtering for correctness and translation quality, around 9.7k multilingual CoT samples are retained. The model based on Qwen2.5-7B-Instruct is fine-tuned with a learning rate of 1e-5, batch size 128, for 3 epochs. Based on Llama3-8B-Instruct, the learning rate is 9e-7, batch size 64, for 1 epochs.

- **LIDR** (Language Imbalance Driven Rewarding) (Yang et al., 2024c) leverages performance gaps between dominant and underrepresented languages as implicit preference signals. LIDR applies DPO training on constructed preference bilingual CoT pairs in 10 languages align to our training data. Based on Qwen2.5-7B-Instruct and Llama3-8B-Instruct, we used 8.9K or 6.4k preference pairs data to train the LIDR model, respectively. The learning rate is 9e-7, batch size is 64, and epoch is 1 for both of them.

- **MAPO** (Multilingual-Alignment-as-Preference Optimization) (She et al., 2024) aligns reasoning across languages using translation-based alignment scores. Following the original setup, we fine-tune using DPO with a learning rate of 1e-6, batch size 128, up to 1,000 steps, and select the best checkpoint based on validation loss.

## D   PER-LANGUAGE RESULTS

The per-language results on four benchmarks are shown in Table 5, Table 6, Table 7, Table 8, and Table 9.

Table 5: The per-language results in MGSM benchmark.

| MGSM | AVG | HRL | URL | EN | DE | FR | ES | RU | ZH | JA | TH | TE | BN | SW |
|------|-----|-----|-----|----|----|----|----|----|----|----|----|----|----|----|
| *Qwen2.5-7B-Instruct* | | | | | | | | | | | | | | |
| Base | 67.2 | 75.7 | 48.8 | 89.2 | 73.6 | 74.8 | 79.2 | 78.8 | 80.0 | 68.0 | 73.6 | 36.4 | 68.0 | 17.2 |
| xRFT | 68.5 | 81.1 | 43.2 | 94.0 | 81.6 | 78.0 | 84.4 | 83.6 | 85.2 | 73.6 | 69.6 | 25.2 | 54.4 | 23.6 |
| LIDR | 69.6 | 79.1 | 50.3 | 90.0 | 79.6 | 76.8 | 82.8 | 82.4 | 82.4 | 70.4 | 76.4 | 39.2 | 69.6 | 16.0 |
| MAPO | 66.3 | 75.8 | 47.4 | 84.8 | 78.4 | 76.4 | 79.2 | 78.8 | 74.8 | 67.2 | 74.4 | 33.2 | 62.8 | 19.2 |
| GRPO | 73.0 | 81.1 | 56.6 | 90.4 | 82.8 | 80.0 | 83.6 | 83.2 | 82.4 | 74.4 | 77.6 | 40.8 | 72.0 | 36.0 |
| mGRPO | **75.9** | **84.4** | **58.7** | 94.0 | 86.0 | 83.2 | 88.8 | 86.4 | 84.8 | 77.2 | 81.2 | 42.8 | 74.8 | 36.0 |
| *Llama3-8B-Instruct* | | | | | | | | | | | | | | |
| Base | 52.2 | 57.3 | 37.7 | 79.6 | 59.6 | 60.8 | 63.6 | 59.2 | 57.6 | 43.2 | 48.4 | 26.0 | 41.2 | 35.2 |
| xRFT | 53.9 | 58.4 | 42.3 | 73.2 | 58.0 | 62.4 | 64.0 | 64.8 | 50.8 | 50.4 | 52.4 | 38.8 | 45.2 | 32.8 |
| LIDR | 55.5 | 58.5 | 45.1 | 79.6 | 66.4 | 62.4 | 61.6 | 61.2 | 52.8 | 46.4 | 56.4 | 43.6 | 35.2 | 45.2 |
| MAPO | 60.7 | 63.9 | 50.9 | 80.4 | 66.8 | 65.2 | 67.2 | 65.6 | 60.4 | 58.4 | 63.6 | 44.8 | 53.2 | 42.0 |
| GRPO | 64.6 | 68.8 | 54.2 | 80.8 | 72.0 | 72.4 | 72.4 | 70.0 | 64.4 | 61.6 | 62.8 | 48.4 | 57.6 | 48.0 |
| mGRPO | **68.1** | **72.3** | **58.3** | 82.0 | 74.8 | 75.2 | 80.0 | 74.8 | 65.6 | 63.6 | 63.2 | 51.2 | 64.8 | 54.0 |

## E   ABLATION STUDY

The details results of our ablation study is shown in Table 10.

Table 6: The per-language results in MATH500 benchmark and its translation version in other 6 target languages.

| MATH-500 | AVG | HRL | URL | EN | IT | JA | TR | ZH | TE | SW |
|---|---|---|---|---|---|---|---|---|---|---|
| *Qwen2.5-7B-Instruct* | | | | | | | | | | |
| Base | 57.9 | 60.9 | 45.5 | 70.8 | 68.4 | 61.6 | 51.8 | 61.6 | 52.4 | 38.6 |
| xRFT | 53.1 | 59.0 | 31.3 | 73.4 | 63.0 | 59.0 | 54.0 | 59.8 | 32.8 | 29.8 |
| LIDR | 62.5 | 66.7 | 48.6 | 73.2 | 70.6 | **68.4** | 65.6 | 62.2 | 57.2 | 40.0 |
| MAPO | 61.2 | 64.6 | 46.9 | 76.2 | 68.4 | 63.0 | 58.6 | 68.4 | 54.6 | 39.2 |
| GRPO | 63.2 | 68.0 | 47.8 | 74.8 | 69.6 | **68.4** | 68.4 | 65.6 | 56.2 | 39.4 |
| mGRPO | **64.9** | **69.8** | **49.2** | 76.8 | **71.8** | **68.4** | **70.4** | 68.8 | **57.6** | **40.8** |
| *Llama3-8B-Instruct* | | | | | | | | | | |
| Base | 26.0 | 27.0 | 22.3 | 29.2 | 26.0 | 29.0 | 26.6 | 26.6 | 25.4 | 19.2 |
| xRFT | 23.4 | 24.5 | 19.0 | 27.4 | 26.2 | 26.4 | 20.6 | 25.0 | 21.2 | 16.8 |
| LIDR | 24.3 | 24.5 | 22.1 | 28.0 | 25.8 | 25.4 | 24.2 | 22.6 | **26.0** | 18.2 |
| MAPO | 25.3 | 25.6 | 21.9 | 30.4 | 28.0 | 26.4 | 24.2 | 24.0 | 24.6 | 19.2 |
| GRPO | 24.4 | 25.7 | 19.1 | 30.0 | 28.0 | 24.6 | 24.8 | 25.4 | 19.2 | 19.0 |
| mGRPO | **26.7** | **27.1** | **23.2** | **32.0** | **29.8** | **27.2** | **26.4** | 25.2 | 24.6 | **21.8** |

Table 7: The per-language results in four difficulty levels of PolyMath benchmark based on Qwen2.5-7B-Instruct.

| Model | AVG | EN | ZH | ES | AR | FR | BN | PT | RU | ID | DE | JA | SW | VI | IT | TE | KO | TH | MS |
|---|---|---|---|---|---|---|---|---|---|---|---|---|---|---|---|---|---|---|---|
| PolyMath-Low | | | | | | | | | | | | | | | | | | | |
| Qwen2.5-7B-Instruct | 74.65 | 89.6 | 79.2 | 87.2 | 80.0 | 84.0 | 66.4 | 80.8 | 83.2 | 81.6 | 76.0 | 68.8 | 14.4 | 79.2 | 83.2 | 36.8 | 74.4 | 73.6 | 79.2 |
| xRFT | 63.08 | 83.2 | 72.8 | 70.4 | 66.4 | 68.0 | 50.4 | 68.0 | 68.8 | 62.4 | 65.6 | 64.8 | 10.4 | 68.8 | 68.0 | 20.0 | 67.2 | 54.4 | 64.8 |
| LIDR | 76.43 | 89.6 | 84.8 | 84.8 | 80.8 | 86.4 | 66.4 | 80.0 | 84.0 | 83.2 | 80.8 | 75.2 | 17.6 | 80.0 | 81.6 | 38.4 | 76.8 | 76.8 | 81.6 |
| MAPO | 77.72 | 92.0 | 81.6 | 88.0 | 80.8 | 84.8 | 67.2 | 80.0 | 89.6 | 84.8 | 77.6 | 75.2 | 23.2 | 85.6 | 85.6 | 43.2 | 77.6 | 74.4 | 82.4 |
| GRPO | 79.69 | 92.0 | 84.0 | 86.4 | 87.2 | 82.4 | 71.2 | 85.6 | 88.0 | 87.2 | 80.8 | 75.2 | 32.8 | 83.2 | 87.2 | 44.8 | 81.6 | 76.0 | 82.4 |
| mGRPO | 81.48 | 94.4 | 83.2 | 88.8 | 87.2 | 86.4 | 76.0 | 88.0 | 87.2 | 85.6 | 82.4 | 78.4 | 36.0 | 85.6 | 88.8 | 44.8 | 80.0 | 80.8 | 84.0 |
| PolyMath-Medium | | | | | | | | | | | | | | | | | | | |
| Qwen2.5-7B-Instruct | 24.00 | 26.4 | 20.0 | 24.8 | 21.6 | 29.6 | 25.6 | 23.2 | 27.2 | 26.4 | 27.2 | 20.8 | 14.4 | 24.8 | 26.4 | 20.8 | 25.6 | 18.4 | 23.2 |
| xRFT | 19.20 | 28.8 | 16.0 | 21.6 | 20.0 | 21.6 | 15.2 | 20.0 | 20.8 | 19.2 | 16.8 | 19.2 | 12.8 | 17.6 | 21.6 | 16.0 | 13.6 | 14.4 | 16.0 |
| LIDR | 25.05 | 28.0 | 20.0 | 28.8 | 26.4 | 27.2 | 23.2 | 27.2 | 26.4 | 27.2 | 24.0 | 22.4 | 18.4 | 26.4 | 29.6 | 16.8 | 25.6 | 28.0 | 25.6 |
| MAPO | 23.02 | 32.0 | 23.2 | 24.8 | 23.2 | 22.4 | 2.4 | 21.6 | 29.6 | 23.2 | 27.2 | 24.0 | 17.6 | 28.0 | 25.6 | 24.0 | 20.8 | 23.2 | 25.6 |
| GRPO | 23.88 | 26.4 | 23.2 | 26.4 | 24.0 | 22.4 | 18.4 | 25.6 | 28.8 | 24.8 | 24.0 | 22.4 | 17.6 | 26.4 | 27.2 | 18.4 | 24.0 | 21.6 | 20.0 |
| mGRPO | 25.97 | 29.6 | 31.2 | 26.4 | 31.2 | 24.8 | 22.4 | 28.0 | 26.4 | 24.0 | 25.6 | 23.2 | 21.6 | 23.2 | 27.2 | 20.0 | 26.4 | 22.4 | 20.8 |
| PolyMath-High | | | | | | | | | | | | | | | | | | | |
| Qwen2.5-7B-Instruct | 9.05 | 8.8 | 7.2 | 9.6 | 8.8 | 6.4 | 10.4 | 10.4 | 10.4 | 9.6 | 14.4 | 5.6 | 3.2 | 12.8 | 10.4 | 6.4 | 8.8 | 10.4 | 8.8 |
| xRFT | 7.88 | 9.6 | 8.0 | 12.0 | 8.0 | 8.8 | 6.4 | 8.0 | 8.8 | 7.2 | 7.2 | 7.2 | 6.4 | 4.8 | 5.6 | 4.8 | 8.0 | 7.2 | 10.4 |
| LIDR | 10.03 | 9.6 | 11.2 | 10.4 | 9.6 | 10.4 | 8.8 | 11.2 | 11.2 | 8.0 | 13.6 | 8.0 | 6.4 | 12.0 | 6.4 | 7.2 | 11.2 | 12.0 | 10.4 |
| MAPO | 8.12 | 11.2 | 8.0 | 7.2 | 8.0 | 5.6 | 9.6 | 7.2 | 7.2 | 8.8 | 8.0 | 12.0 | 3.2 | 9.6 | 9.6 | 5.6 | 7.2 | 8.8 | 8.0 |
| GRPO | 8.68 | 8.0 | 8.0 | 8.0 | 8.0 | 9.6 | 8.0 | 8.8 | 12.0 | 8.8 | 9.6 | 8.0 | 7.2 | 8.8 | 11.2 | 6.4 | 9.6 | 9.6 | 9.6 |
| mGRPO | 10.03 | 8.8 | 9.6 | 12.0 | 10.4 | 12.0 | 8.8 | 9.6 | 10.4 | 10.4 | 10.4 | 11.2 | 5.6 | 9.6 | 13.6 | 6.4 | 8.8 | 8.0 | 10.4 |
| PolyMath-Top | | | | | | | | | | | | | | | | | | | |
| Qwen2.5-7B-Instruct | 5.29 | 7.2 | 4.0 | 6.4 | 4.0 | 7.2 | 4.0 | 4.0 | 6.4 | 5.6 | 4.0 | 4.8 | 5.6 | 5.6 | 5.6 | 4.0 | 2.4 | 5.6 | 4.8 |
| xRFT | 7.94 | 5.6 | 6.4 | 5.6 | 8.8 | 4.8 | 11.2 | 7.2 | 7.2 | 5.6 | 10.4 | 9.6 | 14.4 | 6.4 | 5.6 | 12.0 | 9.6 | 3.2 | 5.6 |
| LIDR | 4.74 | 4.8 | 4.0 | 4.8 | 5.6 | 5.6 | 3.2 | 5.6 | 6.4 | 4.0 | 5.6 | 4.8 | 1.6 | 5.6 | 5.6 | 3.2 | 3.2 | 4.0 | 5.6 |
| MAPO | 6.09 | 5.6 | 5.6 | 4.8 | 6.4 | 8.0 | 5.6 | 5.6 | 7.2 | 6.4 | 10.4 | 4.8 | 2.4 | 6.4 | 4.8 | 3.2 | 5.6 | 4.8 | 8.0 |
| GRPO | 6.71 | 8.0 | 6.4 | 7.2 | 7.2 | 5.6 | 7.2 | 8.0 | 7.2 | 8.0 | 8.8 | 4.8 | 1.6 | 7.2 | 8.8 | 3.2 | 2.4 | 4.0 | 8.8 |
| mGRPO | 7.69 | 6.4 | 8.0 | 8.0 | 7.2 | 7.2 | 9.6 | 8.0 | 8.8 | 7.2 | 8.8 | 6.4 | 5.6 | 8.8 | 7.2 | 4.8 | 5.6 | 4.8 | 4.8 |

Table 8: The per-language results in four difficulty levels of PolyMath benchmark based on Llama3-8B-Instruct.

| Model | AVG | EN | ZH | ES | AR | FR | BN | PT | RU | ID | DE | JA | SW | VI | IT | TE | KO | TH | MS |
|---|---|---|---|---|---|---|---|---|---|---|---|---|---|---|---|---|---|---|---|
| **PolyMath-Low** |
| Llama3-8B-Instruct | 61.42 | 73.6 | 54.4 | 65.6 | 59.2 | 69.6 | 52.8 | 67.2 | 62.4 | 68.8 | 60.8 | 57.6 | 39.2 | 67.2 | 69.6 | 38.4 | 59.2 | 56.0 | 66.4 |
| xRFT | 51.26 | 65.6 | 42.4 | 56.0 | 53.6 | 52.0 | 40.0 | 54.4 | 62.4 | 53.6 | 53.6 | 45.6 | 36.0 | 51.2 | 63.2 | 34.4 | 48.8 | 52.0 | 46.4 |
| LIDR | 54.09 | 61.6 | 49.6 | 59.2 | 50.4 | 60.8 | 44.0 | 65.6 | 58.4 | 56.0 | 56.8 | 45.6 | 37.6 | 57.6 | 58.4 | 36.8 | 47.2 | 52.0 | 51.2 |
| MAPO | 59.69 | 72.8 | 58.4 | 68.8 | 57.6 | 56.8 | 52.0 | 69.6 | 61.6 | 62.4 | 64.8 | 52.0 | 39.2 | 60.0 | 67.2 | 41.6 | 59.2 | 59.2 | 56.0 |
| GRPO | 54.95 | 68.0 | 60.0 | 56.0 | 51.2 | 60.8 | 39.2 | 63.2 | 62.4 | 50.4 | 63.2 | 53.6 | 34.4 | 52.0 | 62.4 | 39.2 | 48.8 | 57.6 | 47.2 |
| mGRPO | 67.62 | 78.4 | 71.2 | 72.0 | 65.6 | 68.8 | 57.6 | 74.2 | 70.4 | 69.6 | 72.8 | 61.6 | 52.0 | 64.8 | 74.4 | 48.8 | 64.8 | 63.2 | 66.4 |
| **PolyMath-Medium** |
| Llama3-8B-Instruct | 4.49 | 8.0 | 3.2 | 6.4 | 5.6 | 2.4 | 2.4 | 4.8 | 5.6 | 5.6 | 4.0 | 6.4 | 2.4 | 1.6 | 4.0 | 4.8 | 7.2 | 6.4 | 6.4 |
| xRFT | 5.17 | 6.4 | 8.0 | 7.2 | 4.8 | 3.2 | 2.4 | 5.6 | 5.6 | 4.0 | 5.6 | 3.2 | 3.2 | 8.0 | 7.2 | 0.8 | 4.8 | 3.2 | 5.6 |
| LIDR | 4.98 | 7.2 | 7.2 | 4.0 | 4.8 | 5.6 | 1.6 | 7.2 | 3.2 | 2.4 | 7.2 | 6.4 | 4.0 | 4.0 | 9.6 | 3.2 | 5.6 | 3.2 | 4.0 |
| MAPO | 4.68 | 6.4 | 2.4 | 4.0 | 4.8 | 7.2 | 5.6 | 4.8 | 4.8 | 4.0 | 4.8 | 5.6 | 2.4 | 4.0 | 0.8 | 4.8 | 3.2 | 4.8 | 3.2 |
| GRPO | 5.11 | 8.0 | 2.4 | 7.2 | 5.6 | 6.4 | 2.4 | 6.4 | 2.4 | 8.0 | 4.8 | 3.2 | 3.2 | 6.4 | 4.8 | 4.0 | 4.0 | 3.2 | 3.2 |
| mGRPO | 5.54 | 9.6 | 4.0 | 8.8 | 4.8 | 4.0 | 4.0 | 6.4 | 2.4 | 5.6 | 7.2 | 4.0 | 4.0 | 5.6 | 6.4 | 6.4 | 5.6 | 4.0 | 4.8 |
| **PolyMath-High** |
| Llama3-8B-Instruct | 1.91 | 2.4 | 3.2 | 1.6 | 2.4 | 3.2 | 0.8 | 2.4 | 0.0 | 2.4 | 2.4 | 1.6 | 1.6 | 0.8 | 1.6 | 1.6 | 1.6 | 2.4 | 2.4 |
| xRFT | 2.34 | 3.2 | 1.6 | 1.6 | 3.2 | 3.2 | 1.6 | 1.6 | 1.6 | 1.6 | 4.0 | 3.2 | 1.6 | 2.4 | 4.0 | 2.4 | 1.6 | 2.4 | 3.2 |
| LIDR | 2.03 | 1.6 | 2.4 | 1.6 | 2.4 | 2.4 | 2.4 | 3.2 | 0.0 | 3.2 | 2.4 | 2.4 | 1.6 | 0.8 | 0.8 | 2.4 | 2.4 | 2.4 | 1.6 |
| MAPO | 1.78 | 2.4 | 0.8 | 3.2 | 1.6 | 1.6 | 2.4 | 2.4 | 0.8 | 2.4 | 1.6 | 0.8 | 1.6 | 1.6 | 3.2 | 1.6 | 0.8 | 1.6 | 0.8 |
| GRPO | 2.46 | 4.0 | 4.0 | 4.0 | 2.4 | 0.8 | 2.4 | 0.8 | 3.2 | 1.6 | 2.4 | 4.0 | 0.8 | 1.6 | 2.4 | 1.6 | 4.0 | 1.6 | 2.4 |
| mGRPO | 2.83 | 2.4 | 4.8 | 2.4 | 1.6 | 4.0 | 3.2 | 2.4 | 3.2 | 3.2 | 3.2 | 1.6 | 1.6 | 3.2 | 3.2 | 1.6 | 1.6 | 3.2 | 3.2 |
| **PolyMath-Top** |
| Llama3-8B-Instruct | 3.02 | 2.4 | 4.0 | 1.6 | 2.4 | 3.2 | 3.2 | 1.6 | 2.4 | 1.6 | 3.2 | 3.2 | 4.8 | 5.6 | 2.4 | 0.8 | 0.8 | 1.6 | 0.8 |
| xRFT | 1.85 | 0.8 | 2.4 | 1.6 | 1.6 | 2.4 | 0.8 | 2.4 | 2.4 | 1.6 | 3.2 | 0.8 | 0.8 | 3.2 | 0.8 | 3.2 | 3.2 | 2.4 | 1.6 |
| LIDR | 2.22 | 1.6 | 4.0 | 0.0 | 0.8 | 0.8 | 1.6 | 3.2 | 4.0 | 2.4 | 3.2 | 1.6 | 3.2 | 2.4 | 3.2 | 0.8 | 2.4 | 0.8 | 3.2 |
| MAPO | 2.89 | 4.8 | 3.2 | 3.2 | 1.6 | 4.0 | 1.6 | 2.4 | 2.4 | 4.0 | 1.6 | 1.6 | 4.0 | 3.2 | 3.2 | 0.8 | 4.0 | 2.4 | 1.6 |
| GRPO | 4.18 | 4.8 | 4.8 | 2.4 | 4.8 | 3.2 | 5.6 | 4.0 | 5.6 | 4.8 | 2.4 | 3.2 | 4.0 | 4.8 | 4.8 | 1.6 | 0.0 | 4.0 | 4.0 |
| mGRPO | 3.57 | 2.4 | 3.2 | 4.0 | 3.2 | 3.2 | 4.8 | 5.6 | 3.2 | 3.2 | 5.6 | 2.4 | 3.2 | 2.4 | 4.0 | 0.8 | 1.6 | 5.6 | 3.2 |

Table 9: The per-language results in X-CSQA benchmark.

| Model | AVG | AR | DE | EN | ES | FR | HI | IT | JA | NL | PL | PT | RU | SW | UR | VI | ZH |
|---|---|---|---|---|---|---|---|---|---|---|---|---|---|---|---|---|---|
| *Qwen2.5-7B-Instruct* |
| Base | 54.3 | 53.0 | 56.0 | 77.1 | 62.9 | 59.3 | 43.2 | 59.5 | 50.6 | 58.0 | 53.8 | 63.6 | 52.9 | 25.5 | 36.3 | 56.0 | 60.6 |
| xRFT | 49.3 | 46.9 | 60.5 | 70.3 | 61.0 | 56.8 | 37.2 | 56.1 | 43.4 | 55.1 | 53.4 | 60.3 | 39.3 | 17.3 | 27.3 | 50.1 | 53.0 |
| LIDR | 53.2 | 53.3 | 58.1 | 75.1 | 60.5 | 57.0 | 40.7 | 54.6 | 52.5 | 54.1 | 55.1 | 56.0 | 54.1 | 26.3 | 36.2 | 57.3 | 60.0 |
| MAPO | 50.7 | 48.6 | 49.7 | 77.1 | 61.1 | 55.5 | 39.2 | 55.2 | 45.2 | 51.9 | 49.8 | 54.7 | 50.7 | 25.1 | 33.4 | 56.5 | 57.3 |
| GRPO | 57.1 | 56.4 | 62.4 | 75.5 | 64.1 | 62.2 | 46.1 | 62.5 | 57.1 | 59.8 | 60.8 | 62.6 | 58.2 | 28.3 | 37.8 | 60.1 | 59.8 |
| mGRPO (Ours) | **60.5** | 58.8 | 65.1 | 82.0 | 68.4 | 67.0 | 50.3 | 66.8 | 57.7 | 61.7 | 62.3 | 65.9 | 62.0 | 31.2 | 40.3 | 63.5 | 64.4 |
| *Llama3-8B-Instruct* |
| Base | 45.1 | 41.7 | 49.7 | 66.3 | 50.6 | 50.6 | 36.8 | 48.1 | 39.0 | 46.6 | 42.0 | 49.3 | 45.7 | 31.4 | 32.5 | 45.8 | 45.8 |
| xRFT | 48.2 | 46.2 | 51.8 | 67.7 | 54.9 | 54.3 | 40.7 | 50.4 | 42.2 | 49.7 | 46.4 | 52.0 | 45.9 | 33.1 | 35.2 | 48.3 | 47.3 |
| LIDR | 52.2 | 47.0 | 55.5 | 69.5 | 57.6 | 55.9 | 46.9 | 55.4 | 47.6 | 52.8 | 51.5 | 55.3 | 54.8 | 38.5 | 41.4 | 52.4 | 53.4 |
| MAPO | 43.9 | 42.4 | 48.9 | 62.3 | 48.1 | 44.7 | 37.5 | 46.6 | 39.0 | 47.0 | 42.1 | 48.8 | 42.7 | 27.8 | 33.2 | 46.0 | 45.2 |
| GRPO | 53.4 | 50.4 | 57.0 | 68.8 | 59.3 | 57.6 | 45.4 | 55.5 | 49.1 | 56.4 | 53.7 | 59.2 | 53.8 | **40.0** | 40.2 | 52.0 | 55.4 |
| mGRPO (Ours) | **53.6** | 50.9 | 57.6 | 70.9 | 60.2 | 58.2 | 47.0 | 57.1 | 50.1 | 55.0 | 52.1 | 57.1 | 54.1 | 39.1 | 42.1 | 54.5 | 52.3 |

Table 10: The results of Ablation Study on MGSM. Best in **bold**.

| MGSM | AVG | HRL | URL | EN | DE | FR | ES | RU | ZH | JA | TH | TE | BN | SW |
|---|---|---|---|---|---|---|---|---|---|---|---|---|---|---|
| Qwen2.5-7B-Instruct | 67.2 | 75.7 | 48.8 | 89.2 | 73.6 | 74.8 | 79.2 | 78.8 | 80.0 | 68.0 | 73.6 | 36.4 | 68.0 | 17.2 |
| GRPO | 73.0 | 81.1 | 56.6 | 90.4 | 82.8 | 80.0 | 83.6 | 83.2 | 82.4 | 74.4 | 77.6 | 40.8 | 72.0 | **36.0** |
| mGRPO | **75.9** | **84.4** | **58.7** | **94.0** | **86.0** | **83.2** | **88.8** | **86.4** | **84.8** | 77.2 | **81.2** | 42.8 | **74.8** | **36.0** |
| w/o format reward | 71.7 | 79.7 | 55.4 | 88.4 | 78.8 | 79.6 | 83.6 | 82.0 | 79.2 | 75.2 | 80.8 | 41.2 | 66.0 | 33.6 |
| w/o unconstrained response | 75.6 | **84.4** | 58.0 | 92.8 | 84.0 | 81.6 | 86.8 | 85.6 | **85.6** | **82.8** | **81.2** | 43.2 | 74.4 | 33.2 |
| only roll-out unconstrained response | 70.4 | 78.2 | 54.7 | 86.4 | 79.2 | 77.2 | 80.4 | 80.4 | 76.4 | 75.6 | 79.6 | **43.6** | 64.4 | 31.2 |
| Qwen2.5-3B-Instruct | 52.4 | 63.1 | 30.4 | 76.4 | 64.4 | 66.4 | 65.6 | 62.4 | 64.0 | 56.0 | 56.0 | 14.4 | 40.4 | 10.8 |
| GRPO | 61.7 | 72.6 | 39.7 | 84.4 | 74.0 | 73.6 | 79.2 | 71.6 | 73.2 | 64.0 | 68.8 | 21.2 | 53.2 | 15.6 |
| mGRPO | 64.7 | 74.9 | **44.3** | 84.4 | 75.6 | 76.8 | **80.0** | 75.6 | 74.8 | 66.8 | 75.2 | 23.6 | 62.4 | 16.0 |
| w/o format reward | 61.6 | 71.3 | 42.1 | 82.0 | 76.4 | 69.2 | 73.2 | 73.6 | 70.8 | 64.4 | 65.2 | 23.2 | 60.0 | 20.0 |
| w/o unconstrained response | 59.3 | 69.2 | 38.8 | 81.6 | 73.2 | 72.0 | 71.2 | 70.0 | 69.6 | 59.2 | 64.0 | 22.4 | 54.4 | 14.4 |
| only roll-out unconstrained response | **64.9** | **76.0** | 42.7 | **86.8** | **78.8** | **77.6** | 79.6 | **78.4** | **76.0** | 65.6 | 70.8 | 21.6 | 57.6 | **20.8** |
| Qwen2.5-1.5B-Instruct | 26.1 | 31.5 | 14.1 | 41.2 | 24.4 | 29.6 | 34.4 | 35.2 | 40.0 | 25.6 | 29.2 | 6.4 | 17.6 | 3.2 |
| GRPO | 47.2 | 58.9 | 23.5 | 72.0 | **63.6** | 60.0 | 62.0 | 62.8 | 58.4 | 46.4 | 47.2 | 8.8 | 32.0 | 6.0 |
| mGRPO | **51.0** | **62.5** | **26.9** | **78.4** | 61.6 | **65.6** | 66.8 | 66.0 | 65.6 | 49.6 | **53.6** | 12.8 | 34.0 | 7.2 |
| w/o format reward | 14.0 | 16.9 | 7.5 | 23.2 | 17.2 | 15.2 | 15.2 | 16.4 | 24.0 | 13.2 | 12.4 | 4.8 | 8.0 | 4.8 |
| w/o unconstrained response | 47.5 | 57.7 | 25.1 | 76.0 | 58.4 | 58.8 | 64.8 | 60.8 | 56.4 | 46.8 | 50.0 | 11.6 | 28.4 | **10.4** |
| only roll-out unconstrained response | 47.9 | 59.7 | 23.7 | 73.6 | 59.2 | 62.0 | 64.0 | 62.4 | 62.0 | 48.4 | 47.6 | 10.4 | 29.2 | 7.6 |

## F  THE NUMBER OF LANGUAGE SETS

To assess the effect of language diversity, we expand it to 15 by adding Arabic (AR), Korean (KO), Portuguese (PT), Telugu (TE), and Vietnamese (VI). We also evaluate reduced settings by randomly selecting 5 languages from the original set, repeating this process three times to assess stability. All experiments are conducted on Qwen2.5-1.5B-Instruct and evaluated on MGSM. Results are shown in Table 11, the 15-language setup slightly hurts overall performance. Reducing to 5 languages leads to further degradation and high variance depending on language selection. These findings indicate that the original 10-language configuration offers a good trade-off between diversity and stability of language sets.

Table 11: Effects of different number of language sets on MGSM with Qwen2.5-1.5B-Instruct.

| MGSM | AVG | HRL | URL | EN | DE | FR | ES | RU | ZH | JA | TH | TE | BN | SW |
|---|---|---|---|---|---|---|---|---|---|---|---|---|---|---|
| Qwen2.5-1.5B-Instruct | 26.1 | 31.5 | 14.1 | 41.2 | 24.4 | 29.6 | 34.4 | 35.2 | 40.0 | 25.6 | 29.2 | 6.4 | 17.6 | 3.2 |
| mGRPO | **51.0** | **62.5** | **26.9** | **78.4** | **61.6** | **65.6** | **66.8** | **66.0** | **65.6** | 49.6 | 53.6 | **12.8** | **34.0** | 7.2 |
| lang_num=15 | 48.7 | 60.5 | 24.7 | 73.2 | 59.6 | 64.0 | 64.4 | 64.0 | 64.4 | 46.8 | **54.0** | 9.6 | 28.4 | 6.8 |
| lang_num=5, (DE, EN, ES, RU, SW) | 47.6 | 59.0 | 24.8 | 70.0 | 59.2 | 62.0 | 61.6 | 58.4 | 63.2 | **49.6** | 50.0 | 8.4 | 33.6 | **7.2** |
| lang_num=5, (ES, FR, SW, TH, ZH) | 44.0 | 54.2 | 22.6 | 68.4 | 56.0 | 55.2 | 58.4 | 57.2 | 54.8 | 43.6 | 44.8 | 10.4 | 29.6 | 5.6 |
| lang_num=5, (ES, FR, RU, SW, ZH) | 15.6 | 20.1 | 6.6 | 25.2 | 18.0 | 20.8 | 16.0 | 17.6 | 30.4 | 17.6 | 10.0 | 2.4 | 7.6 | 6.4 |

## G  ROLL-OUT NUMBER

We also study the effect of varying the roll-out number $n \in \{4, 8, 10, 16\}$ in 1.5B model, and results shown in the top of Table 12. The best performance is observed with $n = 4$. Increasing $n$ to 8 already causes a noticeable drop in performance then ours. With $n = 10$, training becomes unstable due to overexposure to low-resource languages, and we observe a significant amount of garbled text in the model's outputs during later training stages. At $n = 16$, duplicate sampling mitigates some instability. To test whether $n = 4$ generalizes to other model sizes, we also evaluate $n = 4$ on the 3B and 7B models in the Table 12 and lower than $n = 5$.

Table 12: Performance comparison when using different roll-out number (n) in our mGRPO based on three Qwen2.5-Instruct models.

| MGSM | AVG | HRL | URL | EN | DE | FR | ES | RU | ZH | JA | TH | TE | BN | SW |
|---|---|---|---|---|---|---|---|---|---|---|---|---|---|---|
| **Qwen2.5-1.5B-Instruct** | 26.1 | 31.5 | 14.1 | 41.2 | 24.4 | 29.6 | 34.4 | 35.2 | 40.0 | 25.6 | 29.2 | 6.4 | 17.6 | 3.2 |
| n=4 | **52.0** | 62.3 | **29.3** | **80.8** | 60.8 | 64.8 | **68.0** | 60.8 | **67.2** | 52.4 | **56.4** | **16.0** | **35.6** | 9.2 |
| n=5 | 51.0 | **62.5** | 26.9 | 78.4 | **61.6** | **65.6** | 66.8 | **66.0** | 65.6 | 49.6 | 53.6 | 12.8 | 34.0 | 7.2 |
| n=8 | 48.3 | 59.3 | 24.8 | 76.8 | 60.8 | 60.4 | 62.4 | 62.0 | 63.2 | 46.8 | 47.6 | 9.2 | 34.0 | 8.4 |
| n=10 | 15.2 | 19.1 | 8.5 | 18.8 | 17.6 | 17.2 | 18.8 | 20.4 | 25.2 | 15.6 | 16.0 | 5.2 | 7.6 | 5.2 |
| n=16 | 39.9 | 49.9 | 19.3 | 62.4 | 48.4 | 53.2 | 50.8 | 53.6 | 50.8 | 42.8 | 38.0 | 8.4 | 23.6 | 7.2 |
| **Qwen2.5-3B-Instruct** | AVG | HRL | URL | EN | DE | FR | ES | RU | ZH | JA | TH | TE | BN | SW |
| n=5 | **64.7** | **74.9** | **44.3** | **84.4** | 75.6 | **76.8** | **80.0** | 75.6 | **74.8** | **66.8** | **75.2** | **23.6** | **62.4** | **16.0** |
| n=4 | 59.4 | 69.5 | 38.8 | 81.2 | **75.6** | 72.0 | 70.4 | 70.8 | 68.0 | 60.4 | 67.2 | 19.6 | 54.0 | 14.4 |
| **Qwen2.5-7B-Instruct** | AVG | HRL | URL | EN | DE | FR | ES | RU | ZH | JA | TH | TE | BN | SW |
| n=5 | **75.9** | **84.4** | **58.7** | **94.0** | **86.0** | **83.2** | **88.8** | **86.4** | **84.8** | 77.2 | **81.2** | **42.8** | **74.8** | **36.0** |
| n=4 | 73.5 | 82.1 | 56.3 | 90.8 | 84.0 | 81.2 | 85.6 | 83.6 | 80.4 | **78.0** | 79.2 | 39.2 | 72.0 | 34.8 |

## H  MULTILINGUAL THINKING DURING TRAINING

To investigate how the model adheres to "multilingual thinking" prompts during training, we track language consistency ("0" or "1" score) throughout the training process. In the unconstrained setting, we set the language consistency to "1" by default. In preliminary experiments with mGRPO, we observed that in later epochs, the model gradually shifts toward generating English-only reasoning—effectively converging to a behavior similar to GRPO. To closely examine this transition and its impact on performance, we extended training from the originally planned 5 epochs to 10 epochs (700 steps total), using the Qwen2.5-7B-Instruct model as the base.

## H.1 LANGUAGE CONSISTENCY OF PRGM

As shown in Figure 7, GRPO exhibits dominance of English output: while a small amount of non-English (e.g., Chinese, German, Swahili) responses appear initially, the model quickly converges to English-only reasoning. During training, mGRPO exhibits a gradual decline in language consistency from an initially high level, allowing ample room for optimization through multilingual thinking. By epoch 5, the model shifts to generating reasoning almost exclusively in English, indicating a multilingual induction phase followed by a stable, English-dominant regime.

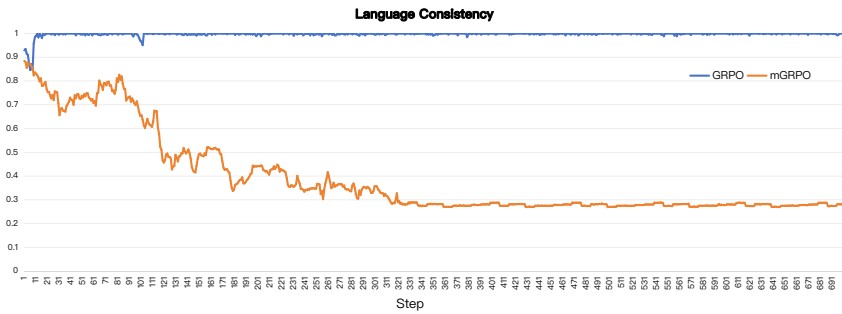

Figure 7: The language consistency of GRPO and mGRPO during training process based on Qwen2.5-7B-Instruct model with 10 epochs (700 steps total).

Under the unconstrained setting, the model initially generates mixed-language responses (e.g., 47.18% English, 42.17% Chinese in epoch 1), rapidly shifting to English-dominant output by epoch 2 (82.70%) and near-exclusively English thereafter. A fine-grained analysis on Thai and Bengali (Table 13) reveals similar dynamics: the base model exhibits multilingual thinking (e.g., Chinese/Thai for Thai questions; English/Chinese for Bengali) and code-switching, while mGRPO transitions from multilingual (base and epoch 1) to stable English-only reasoning. It shows a consolidation process of multilingual exploration into a unified, English-centric reasoning strategy.

Table 13: The response languages for TH and BN on training data with unconstrained-languages prompt.

| MGSM | TH response languages | BN response languages |
|---|---|---|
| Qwen2.5-7B-Instruct | zh: 59.18% th: 40.82% | en: 90.01% zh: 8.39% bn: 1.6% |
| mGRPO    epoch=1 | zh: 82.38% en: 17.20% | zh: 67.73% en: 31.90% |
| epoch=2 | zh: 28.13% en: 71.81% | zh: 2.71% en: 96.87% |
| epoch=3 | en: 99.70% | en: 99.82% |
| epoch=4 | en: 99.76% | en: 99.88% |
| epoch=5 | en: 99.76% | en: 99.70% |
| epoch=6 | en: 99.64% | en: 99.94% |
| epoch=7 | en: 99.76% | en: 99.82% |
| epoch=8 | en: 99.68% | en: 99.82% |
| epoch=9 | en: 99.94% | en: 100.00% |
| epoch=10 | en: 99.82% | en: 99.88% |

Indeed, our evaluation is conducted under unconstrained-language prompts. Although the dominant reasoning language is English, code-switching still occurs in the final responses, e.g., inserting terms or entity from question-language. This demonstrates that, **even when outputting primarily in English, mGRPO retains and leverages a flexible multilingual thinking space at test time**.

## H.2 PERFORMANCE TREND WITH 10 EPOCHS

Figure 8 illustrates the performance evolution on the MGSM benchmark. GRPO reaches near-peak accuracy after just one epoch and quickly plateaus. For mGRPO, performance on HRL (High-Resource Languages) rises sharply in the first epoch and maintains strong growth thereafter. On URL (Under-resourced Languages), mGRPO exhibits a growth trend similar to GRPO during the first

three epochs; however, it continues to improve up to epoch 5, while GRPO has already saturated and struggles to gain further. Notably, even though the training dynamics of mGRPO gradually converge to those of GRPO after epoch 5 (e.g., predominantly English-based reasoning), its performance advantage persists. **It indicates that multilingual thinking in early training establishes a stronger foundation and effectively enhances the model's reasoning capability.** This aligns perfectly with our motivation.

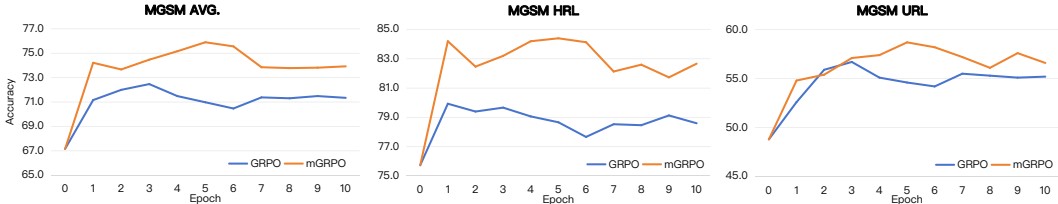

Figure 8: The performance of GRPO and mGRPO in MGSM benchmark with 10 epochs checkpoints trained based on Qwen2.5-7B-Instruct model.

## H.3  EXPLAINING PERFORMANCE DYNAMICS FROM THE PERSPECTIVE OF ENTROPY

We further analyze training dynamics via the entropy of the policy distribution over actions (token-level decisions). As shown in Figure 9, mGRPO starts with significantly higher entropy than GRPO, reflecting greater stochasticity and exploration—likely attributable to the diverse multilingual thinking trajectories. Over time, the entropy of mGRPO steadily decreases and stabilizes at a level lower than that of GRPO, indicating its policy becomes more confident and achieves optimal overall performance. The subsequent convergence of entropy to a level similar to GRPO is also intuitive, as mGRPO increasingly relies on English for reasoning in later stages, aligning its training dynamics with those of GRPO.

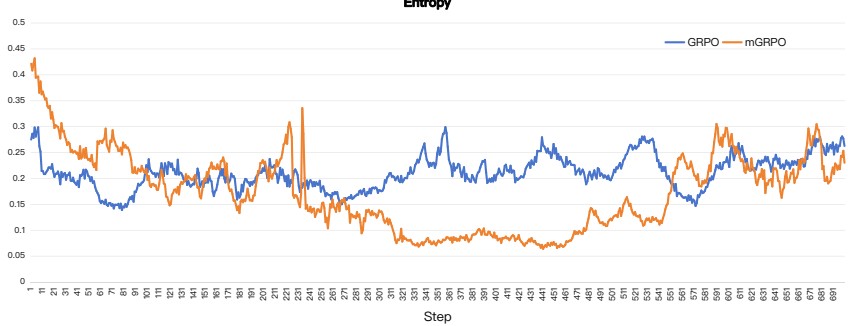

Figure 9: The entropy of GRPO and mGRPO during training process based on Qwen2.5-7B-Instruct model with 10 epochs (700 steps total).

## H.4  CONCLUSION

Therefore, compared to GRPO, mGRPO introduces a beneficial inductive bias via multilingual thinking:

- It fosters broader exploration in early training (higher entropy), leading to richer policy learning;
- It achieves superior final performance, particularly in cross-lingual generalization (evident in HRL and URL subsets);
- Despite eventual convergence toward English-dominated reasoning—a likely artifact of data imbalance or tokenization bias—the intermediate multilingual thinking phase plays a critical role in guiding optimization toward a better basin.

This supports our hypothesis: explicitly encouraging multilingual internal thinking—even if transient—enhances the model's capacity to learn robust, generalizable multilingual policies.

**Future Work** This work represents only a preliminary exploration of multilingual thinking, with future exploration needed on richer multilingual signals—e.g., cross-lingual logical consistency and language-specific reasoning traits.

## I    HOW TO GENERATE A RESPONSE LANGUAGE CONSISTENT WITH THE QUERY LANGUAGE?

Our method focuses on enhancing reasoning capabilities through multilingual thinking, which has shown promising results on both mathematical and commonsense reasoning benchmarks. However, the reasoning process itself predominantly converges toward English, even when the questions are in other languages. This reliance on English limits the direct applicability of the model in user-facing multilingual scenarios.

**We explore a simple test-time strategy that enables mGRPO to achieve a significantly higher language accuracy with only a minor performance trade-off.** We experiment with prepending language-specific prefixes (e.g., "Okay," for English, "D'accord," for French, "Sawa," for Swahili) after the input to guide the model reason in user language. The user language is identified with langid Tool. Besides accuracy, we add a language consistency score to measure whether the generated reasoning matches the query language. The results in MGSM is shown in Table 14. With these prefix, mGRPO obtain 100% language consistency in 10 languages expect low-resource Swahili, while still outperforming GRPO.

Table 14: The results of **Accuracy** and **Language Consistency** on MGSM with language control by language-specific prefix during inference.

| Model | Accuracy | | | | | | | | | | | | | |
|---|---|---|---|---|---|---|---|---|---|---|---|---|---|---|
| | AVG | HRL | URL | EN | DE | FR | ES | RU | ZH | JA | TH | TE | BN | SW |
| Qwen2.5-7B-Instruct | 67.2 | 75.7 | 48.8 | 89.2 | 73.6 | 74.8 | 79.2 | 78.8 | 80.0 | 68.0 | 73.6 | 36.4 | 68.0 | 17.2 |
| w/ prefix | 66.8 | 76.5 | 46.4 | 90.0 | 75.2 | 74.8 | 81.2 | 79.6 | 80.8 | 67.2 | 74.4 | 33.6 | 61.2 | 16.4 |
| GRPO | 73.0 | 81.1 | 56.6 | 90.4 | 82.8 | 80.0 | 83.6 | 83.2 | 82.4 | 74.4 | 77.6 | 40.8 | 72.0 | 36.0 |
| w/ prefix | 72.4 | 82.4 | 52.7 | 91.6 | 83.6 | 78.8 | 86.4 | 85.2 | **85.2** | 75.2 | 80.4 | 35.2 | 70.8 | 24.4 |
| mGRPO | **75.9** | **84.4** | **58.7** | **94.0** | **86.0** | 83.2 | **88.8** | **86.4** | 84.8 | **77.2** | **81.2** | **42.8** | **74.8** | **36.0** |
| w/ prefix | 74.3 | 83.7 | 55.7 | 92.4 | 83.6 | **83.2** | 88.0 | 85.2 | **85.2** | 76.8 | 79.2 | 38.4 | 70.8 | 34.4 |

| Model | Language Consistency | | | | | | | | | | | | | |
|---|---|---|---|---|---|---|---|---|---|---|---|---|---|---|
| | AVG | HRL | URL | EN | DE | FR | ES | RU | ZH | JA | TH | TE | BN | SW |
| Qwen2.5-7B-Instruct | 68.4 | 92.6 | 24.3 | 100.0 | 80.0 | 94.8 | 98.0 | 95.2 | 100.0 | 87.6 | 38.8 | 16.0 | 2.4 | 40.0 |
| w/ prefix | 99.7 | 99.9 | 99.6 | 99.6 | 100.0 | 100.0 | 100.0 | 99.2 | 100.0 | 100.0 | 99.2 | 99.6 | 99.6 | 100.0 |
| GRPO | 17.8 | 14.4 | 2.3 | 100.0 | 1.6 | 3.2 | 1.6 | 0.4 | 74.4 | 5.2 | 1.2 | 0.4 | 1.2 | 6.4 |
| w/ prefix | 99.7 | 99.9 | 99.4 | 100.0 | 99.6 | 100.0 | 100.0 | 100.0 | 100.0 | 100.0 | 100.0 | 100.0 | 99.6 | 98.0 |
| mGRPO | 9.6 | 0.2 | 1.1 | 99.6 | 0.0 | 0.4 | 0.0 | 0.0 | 0.4 | 0.4 | 0.4 | 2.4 | 0.0 | 1.6 |
| w/ prefix | 99.1 | 100.0 | 97.4 | 100.0 | 100.0 | 100.0 | 100.0 | 100.0 | 100.0 | 100.0 | 100.0 | 100.0 | 100.0 | 89.6 |

To enable the model to perform reasoning in the input language, we also attempt a new version of mGRPO, named **mGRPO**$_{lang}$. This version introduces two main modifications: first, all prompts in the PRGM module are constrained response-language; second, a language consistency reward is added to the reward module as a language control signal, as mentioned in GRPO (DeepSeek-AI et al., 2025). We use the FastText (Joulin et al., 2016; Grave et al., 2018) to detect the language of the generated reasoning. When the generated language matches the prompt language, the reward is set to 1; otherwise, it is 0. We train mGRPO$_{lang}$ on the Qwen2.5-7B-Instruct model, keeping all other training parameters the same as before. Evaluation is conducted mainly on the MGSM dataset, with both unconstrained and language-constrained prompts. The results, shown in the Table 15, although mGRPO$_{lang}$ achieves better language consistency across most languages, it experiments a drop in accuracy, especially in low-resource languages.

We present only a preliminary investigation into the language consistency reward, which requires careful design. In particular, both the magnitude and the granularity (e.g., token- vs. sequence-level) of the reward may significantly influence the model's attention to linguistic alignment. For instance, DeepSeek-AI et al. (2025) define the reward as the proportion of tokens conforming to the target language at the token level. Magistral (Rastogi et al., 2025) achieves notable gains in

Table 15: The results of **Accuracy** on MGSM and the **Language Consistency** between query and response languages.

| Model | Accuracy | | | | | | | | | | | | | |
|---|---|---|---|---|---|---|---|---|---|---|---|---|---|---|
| | AVG | HRL | URL | EN | DE | FR | ES | RU | ZH | JA | TH | TE | BN | SW |
| Qwen2.5-7B-Instruct | 67.2 | 75.7 | 48.8 | 89.2 | 73.6 | 74.8 | 79.2 | 78.8 | 80.0 | 68.0 | 73.6 | 36.4 | 68.0 | 17.2 |
| mGRPO | **75.9** | **84.4** | **58.7** | **94.0** | 86.0 | **83.2** | **88.8** | **86.4** | 84.8 | **77.2** | **81.2** | 42.8 | **74.8** | 36.0 |
| mGRPO$_{lang}$ w/ unconstrained prompt | 74.4 | 83.4 | 56.0 | 93.6 | **86.8** | 81.2 | 88.4 | 85.2 | **85.6** | 73.2 | 79.6 | 44.8 | 72.8 | **26.8** |
| mGRPO$_{lang}$ w/ language-constrained prompt | 66.0 | 78.1 | 42.3 | 87.6 | 78.8 | 78.0 | 80.8 | 81.2 | 76.0 | 74.0 | 77.2 | 12.8 | 61.6 | 17.6 |

| Model | Language Consistency | | | | | | | | | | | | | |
|---|---|---|---|---|---|---|---|---|---|---|---|---|---|---|
| | AVG | HRL | URL | EN | DE | FR | ES | RU | ZH | JA | TH | TE | BN | SW |
| Qwen2.5-7B-Instruct | 68.4 | 92.6 | 24.3 | 100.0 | 80.0 | 94.8 | 98.0 | 95.2 | 100.0 | 87.6 | 38.8 | 16.0 | 2.4 | 40.0 |
| xRFT | 95.8 | 99.4 | 89.4 | 100.0 | 99.6 | 100.0 | 99.6 | 98.0 | 100.0 | 99.2 | 95.2 | 96.8 | 74.4 | **91.2** |
| LIDR | 52.3 | 69.1 | 15.2 | 100.0 | 27.6 | 72.8 | 98.0 | 80.4 | 98.8 | 37.2 | 11.2 | 1.1 | 0.8 | 47.6 |
| MAPO | 67.4 | 89.7 | 25.8 | 100.0 | 76.4 | 97.2 | 100.0 | 87.6 | 100.0 | 77.2 | 12.0 | 50.8 | 1.6 | 38.8 |
| GRPO | 17.8 | 14.4 | 2.3 | 100.0 | 1.6 | 3.2 | 1.6 | 0.4 | 74.4 | 5.2 | 1.2 | 0.4 | 1.2 | 6.4 |
| mGRPO | 9.6 | 0.2 | 1.1 | 99.6 | 0.0 | 0.4 | 0.0 | 0.0 | 0.4 | 0.4 | 0.4 | 2.4 | 0.0 | 1.6 |
| mGRPO$_{lang}$ w/ unconstrained prompt | 52.3 | 58.4 | 31.1 | 100 | 57.6 | 66.8 | 56.4 | 28.4 | 98.8 | 42.4 | 99.6 | 0.4 | 1.6 | 22.8 |
| mGRPO$_{lang}$ w/ language-constrained prompt | **99.1** | **100.0** | **97.4** | **100.0** | **100.0** | **100.0** | **100.0** | **100.0** | **100.0** | **100.0** | **100.0** | **100.0** | **100.0** | 89.6 |

language consistency—and with minimal performance degradation—by employing a small amount of multilingual data together with a language consistency reward.

**Future Work** The language consistency reward likely requires more fine-grained design—for instance, ensuring that linguistic consistency does not come at the cost of semantic meaningfulness. Moreover, from an interpretability perspective, one could further investigate how language consistency shapes internal representations and reasoning pathways, thereby informing the design of more targeted reward schemes or training curricula.

## J  THE PERFORMANCE ON BASE LLM

Since LIDR and MAPO are evaluated on Instruct models, our main experiments use Instruct models as well. We also experimented on Qwen2.5-7B base models. However, we observed that base models indeed lack strong instruction-following capabilities, often leading to undesirable continuation behaviors, such as generating additional samples (e.g., "### Instruction:" right after "#### final answer").

Table 16: The results in MGSM benchmark based on two base LLM, Qwen2.5-7B and Qwen3-8B.

| Model | AVG | HRL | URL | EN | DE | FR | ES | RU | ZH | JA | TH | TE | BN | SW |
|---|---|---|---|---|---|---|---|---|---|---|---|---|---|---|
| *Qwen2.5-7B* | | | | | | | | | | | | | | |
| Base | 51.45 | 65.27 | 25.00 | 74.40 | 64.0 | 63.2 | 68.8 | 69.2 | 73.2 | 53.2 | 38.8 | 13.2 | 38.8 | 9.2 |
| LIDR | 61.67 | 70.33 | 41.90 | 88.80 | 67.6 | 68.4 | 69.6 | 72.4 | 79.2 | 64.8 | 68.0 | 24.0 | 55.6 | 20.0 |
| MAPO | 61.85 | 72.20 | 40.30 | 86.00 | 68.0 | 71.6 | 79.6 | 76.4 | 75.6 | 62.0 | 65.2 | 24.8 | 51.6 | 19.6 |
| GRPO | 70.98 | 78.67 | 54.60 | 90.40 | 80.4 | 79.6 | 80.8 | 81.2 | 80.4 | 69.6 | 77.6 | 39.6 | 65.2 | 36.0 |
| mGRPO | 74.76 | 82.53 | 58.80 | **92.00** | 82.0 | 82.8 | **85.6** | **87.2** | 84.4 | 73.2 | 83.6 | 42.8 | 70.8 | **38.0** |
| mGRPO w/ R1-format | **76.07** | **83.80** | **61.30** | 88.80 | **82.8** | **83.2** | 83.6 | 86.0 | **88.0** | 79.2 | **84.8** | 46.8 | 75.6 | **38.0** |
| *Qwen3-8B* | | | | | | | | | | | | | | |
| Base | 81.45 | 84.93 | 73.20 | 93.60 | 84.4 | 82.0 | 86.8 | 88.0 | 85.2 | 83.2 | 84.4 | 72.8 | 80.0 | 55.6 |
| mGRPO w/ R1-format | **87.27** | **90.47** | **80.00** | **97.20** | **91.2** | **90.8** | **92.4** | **94.4** | **88.4** | **85.6** | **90.8** | **79.2** | **88.8** | **61.2** |

To address this, we introduced a penalty term in mGRPO's format reward: if the model generates such continuations, we subtract 0.5 from the original format reward. For LIDR and MAPO, we directly removed the continuation content during preference data preparation. The experimental results are shown in the top part of Table 16. mGRPO also obtain the SOTA score in MGSM benchmark trained on the base LLM.

We also conducted mGRPO training on newest Qwen3-8B, directly adopting the R1 format used in its original training. The R1 format is to places the reasoning process within "<think>...<\think>" tags and sets the output format to "\boxed{final answer}". The model trained with R1 format is named mGRPO w/ R1 format. The results are shown in bottom of Table 16 and represents that mGRPO is suited for R1 format output and even obtain better performance; and for stronger base LLM, Qwen3-8B, mGRPO also could obtain improvement of performance.

# K EXTENDING MULTILINGUAL THINKING TO GSPO

Since our method primarily modifies the rollout procedure of GRPO—i.e., sampling reasoning traces in multiple languages—it can be readily adapted to GRPO variants such as DAPO (Yu et al., 2025) and GSPO (Zheng et al., 2025). Notably, GSPO elevates the optimization unit in reinforcement learning from the token level to the entire sequence level. Specifically, it replaces the per-token importance ratio $\frac{\pi_\theta^{i,t}}{\pi_{\theta_{\text{ref}}}^{i,t}}$ with a sequence-level ratio, normalized by sequence length:

$$s_i(\theta) = \left( \frac{\pi_\theta^i(o_i \mid p_i, q)}{\pi_{\theta_{\text{ref}}}^i(o_i \mid p_i, q)} \right)^{\frac{1}{|o_i|}} = \exp\left( \frac{1}{|o_i|} \sum_{t=1}^{|o_i|} \log \frac{\pi^{i,t}(o_{i,t} \mid p_i, q, o_{i,<t})}{\pi_{\theta_{\text{ref}}}^{i,t}(o_{i,t} \mid p_i, q, o_{i,<t})} \right) \quad (6)$$

Additionally, GSPO discards the KL-divergence penalty term used in GRPO. Consequently, the mGSPO objective simplifies to:

$$\mathcal{L}_{\text{mGSPO}}(\theta) = \mathbb{E}_{(q,a)\sim\mathcal{D}, \{o_i\}_{i=1}^n \sim \pi_{\theta_{\text{ref}}}(o_i|p_i,q)_{i=1}^n}$$
$$\left[ \frac{1}{n} \sum_{i=1}^n \left\{ \min\left[ s_i(\theta)\hat{A}_i, \text{clip}\left( s_i(\theta), 1-\epsilon, 1+\epsilon \right) \hat{A}_i \right] \right\} \right] \quad (7)$$

We implement both GSPO and mGSPO based on Qwen2.5-7B-Instruct, using identical data, hyper-parameters, and evaluation protocols as in prior experiments. Results on the MGSM benchmark (Table 17) show that: GSPO underperforms GRPO-based methods—likely due to its sequence-level credit assignment being suboptimal for multi-step reasoning. Nevertheless, mGSPO outperforms GSPO by +6.6% and GRPO by +1.3% in average accuracy, confirming that **multilingual thinking consistently enhances reasoning capability—even under different RL optimization granularities.**

This further validates the robustness and transferability of multilingual thinking as a general inductive bias mentioned in Appendix H.

Table 17: The results in MGSM benchmark based on GSPO and mGSPO.

| MGSM | AVG | HRL | URL | EN | DE | FR | ES | RU | ZH | JA | TH | TE | BN | SW |
|---|---|---|---|---|---|---|---|---|---|---|---|---|---|---|
| *Qwen2.5-7B-Instruct* | | | | | | | | | | | | | | |
| Base | 67.2 | 75.7 | 48.8 | 89.2 | 73.6 | 74.8 | 79.2 | 78.8 | 80.0 | 68.0 | 73.6 | 36.4 | 68.0 | 17.2 |
| xRFT | 68.5 | 81.1 | 43.2 | **94.0** | 81.6 | 78.0 | 84.4 | 83.6 | 85.2 | 73.6 | 69.6 | 25.2 | 54.4 | 23.6 |
| LIDR | 69.6 | 79.1 | 50.3 | 90.0 | 79.6 | 76.8 | 82.8 | 82.4 | 82.4 | 70.4 | 76.4 | 39.2 | 69.6 | 16.0 |
| MAPO | 66.3 | 75.8 | 47.4 | 84.8 | 78.4 | 76.4 | 79.2 | 78.8 | 74.8 | 67.2 | 74.4 | 33.2 | 62.8 | 19.2 |
| GRPO | 73.0 | 81.1 | 56.6 | 90.4 | 82.8 | 80.0 | 83.6 | 83.2 | 82.4 | 74.4 | 77.6 | 40.8 | 72.0 | 36.0 |
| mGRPO | **75.9** | **84.4** | **58.7** | **94.0** | **86.0** | **83.2** | **88.8** | **86.4** | **84.8** | **77.2** | **81.2** | **42.8** | **74.8** | **36.0** |
| GSPO | 67.7 | 76.8 | 49.5 | 85.6 | 76.8 | 73.2 | 81.2 | 78.8 | 80.0 | 70.8 | 74.0 | 30.0 | 65.6 | 28.4 |
| mGSPO | **74.3** | **83.9** | **55.5** | **92.4** | **84.4** | **84.0** | **84.8** | **88.4** | **85.2** | **76.4** | **76.4** | **42.0** | **69.6** | **34.0** |