# OpenReview forum: "mGRPO: Unlocking LLM Reasoning through Multilingual Thinking"
_ICLR.cc/2026/Conference — Submitted to ICLR 2026_

### Official Review · Reviewer_EMqQ · 2025-10-27

**Soundness:** 3
**Presentation:** 2
**Contribution:** 2
**Rating:** 4
**Confidence:** 4

**Summary:**

This paper introduces MGRPO, a framework designed to enhance the reasoning ability of large language models (LLMs) through online multilingual preference data generation. At the core of MGRPO is the Polyglot Reasoning Generation Module, which guides the LLM to generate multilingual reasoning paths to optimize LLM. Experimental results demonstrate that MGRPO consistently improves performance across four reasoning benchmarks and two base models.

**Strengths:**

By explicitly generating multilingual preference groups, including one unconstrained output and several language-constrained outputs, the approach enables the model to leverage a wider and more diverse reasoning space than prior reinforcement learning-based methods, most of which remained English-centric. Building on this insight, this work incorporates multilingual reasoning into the rollout phase to further strengthen the model's reasoning capability.

**Weaknesses:**

1. While the motivation to enhance LLM reasoning via multilingual thinking is commendable, the execution is fundamentally misaligned with this goal. The proposed methodology effectively reduces to cross-lingual thinking, rather than genuine multilingual reasoning.
2. The proposed MGRPO framework appears to operate without an explicit constraint on the reasoning language, particularly under its optimal performance setting. While this language-agnostic approach aligns with a more flexible "multilingual thinking" concept, it introduces an ambiguity: we do not know which language is predominantly utilized for the internal Chain-of-Thought (CoT) generation.

**Questions:**

1. In Figure 2, the specified-language settings are defined in the prompt. However, does the model actually follow these language settings during reasoning?
2. In Section 5.3, this work examined token activations at each decoding step across all layers. The authors could further analyze the actual reasoning languages used by the model under its best-performing conditions for each target language.
3. $\text{MGRPO}_{lang}$ shows a decline in accuracy, particularly for low-resource languages.
Why does adding the language consistency reward lead to worse performance compared to the base model?


Typos:
1. Line 206, "REWARD MODUL" should be "REWARD MODULE"
2. Line 965, "espicially " should be " especially"
3. Line 965, "low-resour" should be "low-resource"

---

> ### Author Response · Authors · 2025-11-22
> **Response to Reviewer EMqQ**
>
> ### **R1: Response to "Multilingual Thinking Method and Cross-lingual goal misalignment"**
>
> Multilingual thinking and cross-lingual learning are not contradictory. Thinking about the same problem in multiple languages is actually similar to cross-lingual thinking. Our "multilingual thinking" refers to the response itself, not the way the question is answered.
>
> ### **R2: Response to “which language is predominantly utilized for the internal Chain-of-Thought (CoT) generation” and related Q2.**
>
>  Under unconstrained decoding, we observe that **English tokens dominate across most layers**—suggesting a strong internal preference for English-based reasoning. Intriguingly, when we apply language-specific prefixes (as in Appendix H of the original submission) to encourage language-aligned output, token usage evolves as follows:
> - In the first three quarters of the network depth, English tokens remain predominant;
> - In the final quarter of layers, tokens in the target language progressively increase, culminating in predominantly non-English tokens at the output layer.
>
> This layer-wise pattern suggests that *multilingual thinking* doesn’t mean the model literally “thinks” in many languages at the token level. Instead, it uses English as a common internal reasoning framework, while remain language-specific output token.
>
> To further investigate this, we analyze token-level entropy during training (see **Figure 9** in the revised manuscript):
>  - mGRPO exhibits a **significantly higher initial entropy** than GRPO, indicating greater exploration—consistent with the activation of a richer set of reasoning trajectories via multilingual prompting;
>  - Over time, the entropy of mGRPO steadily decreased and stabilized below that of GRPO, reflecting a more confident and stable strategy. Ultimately, English dominated the output tokens, likely due to the underlying LLM model having stronger prior knowledge of English due to bias in the pre-training data.
>
> In summary, these findings support the idea that *multilingual thinking* enhances reasoning ability not by forcibly generating superficial multilingualism, but by expanding the model's internal exploration space during training, ultimately leading to more robust English strategies during reasoning.
>
> For full technical details, please refer to the updated Appendix H.
>
> ### **R3: Response to Q1 about “Languages in Traning process”**
> We track language consistency of reasoning traces throughout training and present the results in **Figure 7** of the revised manuscript. In mGRPO, **language consistency gradually decreases over epochs, and by the final epoch, reasoning predominantly shifts to English**—mirroring the convergence pattern observed in GRPO.
>
> To assess whether prolonged training would cause mGRPO to fully align with GRPO, we extend both methods from 5 to 10 epochs. Results show that while both converge toward English-dominant reasoning, **mGRPO consistently maintains a performance advantage over GRPO throughout all 10 epochs**, confirming that the benefit of multilingual thinking persists beyond early training stages.
>
>  Full experimental details and discussion are provided in the updated **Appendix H**.
>
> ### **R4: Response to Q3 about “Why does adding the language consistency reward lead to worse performance compared to the base model?”**
>
> When evaluating mGRPO_lang with prompts that explicitly request responses in the input language, we observe severe degradation on low-resource languages, most generate notably, repetitive, non-informative outputs (e.g., *“hili hili hili…”* in Swahili). **We also observed during training that while the model adheres to language consistency constraints, it tends to output useless repetitive text in responses from low-resource languages.** This may explain why mGRPO_lang performs worse than the base model in low-resource languages.
>
> To clarify the scope of this issue, we additionally report results under *unconstrained* decoding (i.e., unconstrained response-language specification in the prompt) in **Table 15** (revised manuscript). Here, mGRPO_lang achieves slightly higher language consistency than mGRPO, yet still underperforms GRPO on average of MGSM, though it remains above the base model. **This indicates that the language consistency reward primarily harms performance on low-resource languages, while having milder effects on high-resource ones.**
>
> These findings suggest that future work should refine the design of language consistency rewards—for instance, by jointly enforcing *semantic coherence* and *language consistency*, or by dynamically weighting the reward based on language resource levels. We have expanded on such directions in the **Future Work** discussion (Appendix I, revised manuscript).
>
> ---
>
> I hope my reply has cleared up your confusion, and I look forward to further communication with you.

---

### Official Review · Reviewer_BddF · 2025-10-29

**Soundness:** 2
**Presentation:** 2
**Contribution:** 2
**Rating:** 2
**Confidence:** 5

**Summary:**

This paper introduces mGRPO, an effective method for improving LLM reasoning performance by exposing the model to diverse languages during policy optimization, even if the resulting enhanced reasoning pathway ultimately defaults to generating answers via an internal, English-centric process.

**Strengths:**

The primary strength of the paper is its empirical demonstration that using multilingual training roll-outs within a reinforcement learning framework achieves SOTA accuracy across multiple multilingual reasoning benchmarks.

**Weaknesses:**

1. The paper's central claim is to leverage "multilingual thinking".  However, the analysis section (Appendix H) explicitly reveals that the learned policy predominantly collapses back to English reasoning during inference. For German, French, Spanish, and Russian, the model achieves 0.0% to 0.4% consistency, indicating that regardless of the input language, the model reasons and responds in English. It is totally contradicts the proposed idea that model is performing reasoning in a diverse multilingual space. The English thinking makes the whole framework non-sense where the prompts are think in Swahili/Japanese/Russian. So, the core idea of "multilingual thinking" is not convincingly supported by the experimental evidence.

2. Although the paper added a language consistency reward to encourage the model to think in the input language, the results demonstrate that this reward will be degraded the performance on reasoning tasks. While authors have considered the language unconsistency issue, why the proposed reward function (Section 3.2) not include this part is not well explained.

3. The calculation of the language consistency reward is not the same with the paper which proposed GRPO. This paper assign the reward 1 or 0 based on whether the generated answer is in the same language as the input question. However, GRPO uses a more fine-grained approach by calculating the proportion of tokens in the generated answer that match the input language. It neglects the cases of code-mixing.

4. After using the language consistency reward, the performance on reasoning tasks drops significantly (Table 13).  This drop is particularly severe in the low-resource languages (URL), where the accuracy plummets from 42.8 to 12.8. These results highlight a major flaw, that the only way the method achieves its best performance is by following the reasoning process to default to English, suggesting the core idea 'multilingual thinking' is fundamentally flawed.


5. This method is not robust to different settings, e.g. the sensitivity to the roll-out number n. As shown in Table 12, the best choice of n varies across different models and languages, making it difficult to select a universal n for practical applications.

6. For the experiments, the authors only compare the standard GRPO (English only) and mGRPO (multilingual roll-outs). However, the multilingual training data for mGRPO is generated online from a starting set of 1,703 English questions translated into nine languages (18,140 examples total). It lacks one important baseline where the standard GRPO policy is simply trained on a much larger, but still English-only, dataset of comparable size. This leaves open the possibility that the gains are due more to the quantity of the training data (translated questions acting as diverse reasoning prompts) rather than the mechanism of multilingual thinking policy optimization itself.

**Questions:**

see weaknesses above.

---

> ### Author Response · Authors · 2025-11-22
> **Response to Reviewer BddF (1/2)**
>
> Thank you very much for recognizing the performance of our mGRPO. We will now address your questions one by one.
>
> ---
>
> ### **R1: Response to “The core claim—*multilingual thinking*—lacks strong experimental support, as the model consistently reasons and responds in English regardless of input language.”**
>
> We respectfully address this concern in three points:
>
> 1. **Performance gain is attributable to *multilingual thinking***
>
> Despite predominantly English outputs, the performance gain is directly attributable to *multilingual thinking* during training: replacing it with English-only reasoning (i.e., GRPO) reduces MGSM accuracy by 2.91 points (75.93 → 73.02) on Qwen3 and by 3.53 points (68.11 → 64.58) on Llama3.
>
> 2. **Multilingual thinking is a means to improve reasoning, not the ultimate goal**
>
>  Our primary objective is to enhance reasoning capability, not to enforce multilingual output. Consistent with this, during inference we do not constrain the response language, allowing the model to freely choose its output language, as specified in the *Inference Setup* (Lines 309–311).
>
>  3. **Language consistency responses are feasible with minimal performance cost**
>
> As noted in Lines 467–468 and Appendix H (Lines 967–971), simple test-time language prompting (e.g., *“Okay,”* for English, *“D’accord,”* for French) effectively elicits language-consistent outputs. Table 14 (revised as Table 13 in  new version) shows only a 1.6% average drop on MGSM, with mGRPO still outperforming GRPO by 1.9%—its preference for English output reflects stylistic convention rather than absence of multilingual capability.
>
>  In summary, the empirical evidence robustly supports our core claim: *multilingual thinking enhances LLM reasoning ability*.
>
>  To further strengthen this argument, the revised manuscript includes an in-depth analysis in **Appendix H**, which examines policy entropy dynamics during training. **We observe that *multilingual thinking* induces higher initial entropy, thereby expanding the exploration space in early training stages and facilitating more effective policy learning.** This provides a mechanistic explanation for the observed performance gains and reinforces the validity of our central thesis.
>
> ### **R2: Response to "While authors have considered the language unconsistency issue, why the proposed reward function (Section 3.2) not include this part is not well explained."**
>
> **The “language inconsistency” issue emerged as an *experimental observation* during post-hoc analysis, after achieving our primary goal of enhancing reasoning capability, and thus constitutes a secondary finding rather than a core problem to be addressed**. For this reason, it is appropriately placed in the appendix rather than the main text.
>
> Nevertheless, in Appendix H (Appendix I in the revised manuscript), we explore two approaches to mitigate this issue: (1) adding a language consistency reward (i.e., mGRPO_lang), and (2) a lightweight test-time prompting strategy using language-specific prefixes. **The latter effectively resolves language inconsistency with negligible performance cost, demonstrating that explicit reward engineering is not necessary.**
>
> ### **R3：Response to "The calculation of the language consistency reward is not the same with the paper which proposed GRPO. "**
>
> As mentioned in R2, "language consistency" is not our core issue, so we only tried a global "language consistency reward" scheme. Thank you for your reminder; we have included the fine-grained language reward design of GRPO in the discussion of future work, as reflected in Lines 1132-1133 of the new version.
>
> ### **R4: Response to “The addition of a language consistency loss leads to significant performance degradation on low-resource languages, suggesting that the method only achieves optimal performance when reasoning defaults to English—implying a fundamental flaw in the core idea of *multilingual thinking*.”**
>
>  The performance drop caused by the language consistency reward does not indicate a fundamental flaw in the *multilingual thinking* concept; rather, it offers a more comprehensive perspective on why *multilingual thinking* is effective in mGRPO.
>
> As shown in Appendix H, mGRPO’s *multilingual thinking* induces high initial entropy, promoting diverse exploration; entropy then declines smoothly, leading to a stable, high-performance policy—typically expressed in English. Crucially, its benefit endures: even after 5 extra epochs, mGRPO consistently outperforms GRPO. In contrast, mGRPO_lang’s language-consistency reward suppresses entropy reduction, trapping the model in a high-entropy regime where reasoning satisfies surface language constraints but lacks substantive quality—especially for low-resource languages. Therefore, **the issue lies not with *multilingual thinking* as a concept, but with the rigidity of the reward-based language consistency enforcement mechanism.**

---

> > ### Author Response · Authors · 2025-11-22
> > **Response to Reviewer BddF (2/2)**
> >
> > ### **R5: Response to "This method is not robust to different settings, e.g., the sensitivity to the roll-out number n."**
> >
> > Table 12 shows that on the Qwen2.5-1.5-Instruct model, although the performance of n=5 (the parameter we uniformly adopted) is not as good as n=4 (1% lower), it still achieves a significant improvement (MGSM's performance improved from the base 26.1 to 51.0). The advantage of n=5 is still obvious on other 3B and 7B models. Therefore, **n=5 has been verified to provide effective performance improvements across models of different sizes and should be a universally applicable setting.** Perhaps you can provide more perspectives to explain your concerns and doubts about our parameter settings.
> >
> > ### **R6: Response to "mGRPO and GRPO data are not aligned"**
> >
> > In Section 4.2, we explained that the **GRPO training data is fully aligned with mGRPO, i.e., 18140 multilingual examples**. That is, the only variable between GRPO and mGRPO is whether "multilingual thinking" is used. I'm not sure if the reviewers overlooked this setting, or if we misunderstood your question.
> >
> > ---
> >
> > We look forward to your further reply!

---

> > > ### Comment · Reviewer_BddF · 2025-11-28
> > >
> > > Thank you for the response.
> > >
> > > The response to R4 confirm my inital concern. You state that enforcing language consistency "suppresses entropy reduction" and leads to poor performance. This implicitly admits that the model **cannot reason in low-resource languages** and must default to English to achieve high scores. And the experimental evidence (Appendix H) confirms that the model is performing **English reasoning triggered by multilingual prompts, not thinking in the target language**. If this method requires to ignore the implicit instruction (think in the target language) to get a good performance, **the proposed "multilingual thinking" mechanism is not functioning as the name suggests**. It is essentially a data augmentation method that forces English reasoning pathways to be more robust.
> > >
> > > For the response to R6, there seems to be a misunderstanding of my question regarding the baselines. My concern is that comparing mGRPO (1,703 questions $\times$ 9 languages) against GRPO (18,140 multilingual inputs) does not validate the efficiency of multilingual data vs. monolingual data. The crucial missing baseline is GRPO trained on $\approx$ 18,000 unique English questions.
> > >
> > > While the authors argue that language inconsistency is a "secondary finding." However, **for a paper targred on multilingual capabilities, the inability to control the language of thought without degrading reasoning is a primary limitation**. Therefore, the rebuttal confirms my intial concern, I will keep my score.

---

> > > > ### Author Response · Authors · 2025-11-30
> > > >
> > > > > 1.Thank you for your prompt reply, which has helped us understand the **fundamental misunderstanding** you have regarding our work.
> > > >
> > > > You characterized our paper as “a paper targeted on multilingual capabilities.” However, we have consistently emphasized that **our goal is to enhance LLMs’ reasoning abilities through multilingual thinking** (e.g., Lines 121 and 159). In our framework, “multilingual thinking” serves as a *means* to improve reasoning accuracy—and indeed, we achieve state-of-the-art results, which you acknowledged.
> > > >
> > > > We suspect that our choice of multilingual benchmarks may have led you to assume our primary objective was “multilingual capability.” Even if that were the case, we believe we have successfully demonstrated such capability: **accurately understanding multilingual questions and generating correct answers is itself a valid manifestation of multilingual capability**. Moreover, whether strict language consistency is *necessary* for reasoning remains debatable. For instance, consider the following three works [1-3] —among which MAPO [2] and LIDR [3] are baselines we compared against:
> > > > - [1] aims to improve multilingual reasoning, yet its appendix still shows responses primarily in English;
> > > > - We evaluated the language consistency of the other two methods and found they also fail to produce consistently language-aligned outputs, especially for low-resource languages.
> > > >
> > > > **Therefore, it would be unfair to dismiss our approach because we discussed the “language consistency”.**
> > > >
> > > > [1] Question Translation Training for Better Multilingual Reasoning, Zhu et al., ACL 2024
> > > > [2] MAPO: Advancing Multilingual Reasoning through Multilingual-Alignment-as-Preference Optimization, She et al., ACL 2024
> > > > [3] Language Imbalance Driven Rewarding for Multilingual Self-Improving, Yang et al., arXiv:2410.
> > > >
> > > >
> > > > Furthermore, **we actually propose a simple and effective method to improve language consistency**—namely, using language-specific prefixes to guide the model’s output (see Table 14 in Appendix I of the revised manuscript, corresponding to Table 14 in the submitted version). This effectively addresses the issue. We included results from the less effective “language consistency reward” scheme only to present a comprehensive set of experimental findings. If this inclusion has caused significant misunderstanding, we are happy to remove those results entirely. Additionally, given the strong performance of modern machine translation systems, one could easily translate an English reasoning chain into the target language—a solution that incurs minimal cost. Thus, if pursuing strict language consistency comes at the expense of substantial reasoning accuracy, it is worth questioning whether such consistency should be a primary optimization goal at all.
> > > >
> > > >
> > > > > 2.Your criticism of our “multilingual thinking” mechanism also stems from a misunderstanding.
> > > >
> > > > **The “multilingual thinking” mechanism *is indeed* present during early training (e.g., the first 4 epochs)**; what changes over time is the *proportion* of generations that follow the language instruction, as clearly shown in Figure 7 of Appendix H. In other words, as training progresses, the model internalizes multilingual reasoning capabilities into its English-based thought process. This is evident from the comparison between mGRPO and GRPO—the only difference between them is the presence of multilingual thinking during training. **It would be incorrect to deny the role of the “multilingual thinking” mechanism simply because the model eventually converges to English outputs.**
> > > >
> > > > > 3.Regarding your comment: “missing baseline is GRPO trained on 18,000 unique English questions.”
> > > >
> > > > This proposed baseline differs from our method in two key aspects: (1) training data composition and (2) rollout langauges. Therefore, such a comparison would be invalid. While we agree with you that using multilingual versus monolingual question sets may involve efficiency (or effectively maybe more correctly), this does not undermine the applicability of our approach. Even when all input questions are in English, the model can still generate reasoning chains in multiple languages—as demonstrated in our “Polyglot Thinking Experiment” (Figure 2).
> > > >
> > > > ---
> > > >
> > > > Thank you again for your thoughtful feedback—it has greatly helped us identify potential points of confusion for readers, and we appreciate the opportunity to clarify them.

---

### Official Review · Reviewer_8cad · 2025-11-01

**Soundness:** 3
**Presentation:** 3
**Contribution:** 3
**Rating:** 6
**Confidence:** 2

**Summary:**

This paper investigates the potential of multilingual thinking to enhance the reasoning capabilities of Large Language Models (LLMs). The authors first present a "Polyglot Thinking Experiment" demonstrating that (1) non-English responses can outperform English ones on reasoning tasks, and (2) an unconstrained setting, where the model freely chooses its response language, often yields the best performance. Building on this insight, the paper proposes Multilingual Group Relative Policy Optimization (mGRPO), a reinforcement learning framework. mGRPO adapts the GRPO algorithm by generating a group of responses for each prompt, including one from an unconstrained prompt and several from language-constrained prompts (randomly sampling different languages). This polyglot group is then used to compute group-relative advantages based on a simple reward function combining final answer accuracy and response format. The model is trained online using only 18.1k examples. Experiments on two base models (Qwen2.5-7B, Llama3-8B) across four multilingual reasoning benchmarks (MGSM, MATH500, PolyMath, X-CSQA) show that mGRPO significantly outperforms the base models and strong baselines, including a standard English-only GRPO, demonstrating the effectiveness of leveraging multilingual thinking for improving reasoning.

**Strengths:**

1. The "Polyglot Thinking Experiment" (Sec 1, Fig 2) provides a clear and compelling empirical foundation for the paper's entire premise. The finding that an unconstrained language setting often outperforms both English-only and specific language-constrained settings is a novel and significant observation.
2.	The proposed mGRPO framework is a clever and original adaptation of GRPO. The core idea of constructing a "polyglot reasoning set" for group-wise optimization, mixing unconstrained and language-constrained rollouts, is a direct and principled response to the motivating experiment.
3.	The method is lightweight. It relies on a very small training dataset (~18k examples) and, crucially, does not require supervised chain-of-thought (CoT) data or human-generated preference labels. The reward function is simple, rule-based, and computed online, making the approach highly scalable.

**Weaknesses:**

1. While the unconstrained setting is central to the method's motivation and success, the paper provides limited analysis of what the model actually does in this setting. The text mentions (Sec 1, line 096) that "the model typically reasons in English but borrows surface entities." This is a key detail. During inference (when the [LANGUAGE] token is empty), what languages does the model actually choose to reason in? Does this choice vary by the language of the input question? A more detailed analysis of the unconstrained behavior (e.g., a distribution of languages chosen for CoT during inference) would significantly strengthen the paper's claims about the model "freely choosing" its reasoning path.
2. The simplicity of the reward function ($r \in \{0, 1, 2\}$) is a strength (scalability) but also a potential weakness. It only rewards the final answer and the format. This means a response with a completely flawed reasoning path that luckily arrives at the correct answer (and is formatted correctly) receives the maximum reward (2), identical to a response with a perfectly sound reasoning path. While this "ends-justify-the-means" approach clearly works well, it relies on the model stochastically finding correct paths. The paper would be stronger if it discussed the potential limitations of this reward signal.
3. The paper notes (Sec 5.1, 5.3) that the model's reasoning paths tend to converge to English during training and inference. While the logit lens analysis suggests this is a feature (fusing knowledge), it also raises a question: is the multilingual-thinking-as-scaffolding a temporary effect? If trained for much longer, would mGRPO's performance simply converge to that of the standard GRPO? The paper explores language consistency in Appendix H, but a discussion on the long-term training dynamics and whether the multilingual "scaffolding" benefit is persistent would be valuable.

**Questions:**

1.	Following up on Weakness #1: Could the authors provide a more detailed analysis of the model's behavior during inference in the unconstrained setting? For instance, when given a question in Thai (TH) on the MGSM benchmark, what language does the mGRPO-trained model predominantly use for its [Thinking] steps? Does this differ from the base model? This seems critical to understanding how mGRPO utilizes its "flexible thinking space" at test time.
2.	Regarding the reward function: The format reward FR is binary. Did the authors experiment with a more granular reward, for example, a small reward for just having [Thinking] or just #### [Answer], or a penalty for generating extraneous text (as explored in Appendix I for base models)?
3.	The performance on Llama3-8B (Table 1) shows mGRPO provides a large boost over a weak base model, but the final scores (e.g., 68.11 on MGSM) are still lower than the mGRPO-trained Qwen model (75.93). This suggests the base model's multilingual capability is a strong prerequisite. Could the authors comment on how much "base" multilingualism is necessary for mGRPO to be effective?
4.	The analysis of the roll-out number n in Appendix G (Table 12) is interesting. It seems n=4 or n=5 is optimal, and performance drops with more rollouts (e.g., n=8 or n=10), which is somewhat counter-intuitive. The authors attribute this to "overexposure to low-resource languages." Does this imply that the quality of the non-English rollouts is often very low, and adding too many of them introduces more noise than signal to the group-relative advantage estimation?

---

> ### Author Response · Authors · 2025-11-22
> **Response to Reviewer 8cad (1/2)**
>
> We sincerely thank you for your thorough review of our work; your recognition of the **motivation** and **methodological** design of our approach is greatly encouraging. In response to your constructive suggestions for improvement, we have conducted careful reflection and preliminary explorations, and we address each point below.
>
> ---
>
> ### **R1: Response to “detailed analysis of the model's behavior during inference in the unconstrained setting” and related Q1**
>
> First, **let's clarify a part that was inappropriately described in the original text about unconstrained response-language setting. In Sec 1, line 093**, we mentioned that "the model typically reasons in English but borrows surface entities." This observation mainly occurred in Qwen2.5-1.5B-Instruct and Qwen2.5-1.5B-Instruct, which have weaker multilingual capabilities. However, **in Qwen2.5-7B-Instruct, which has stronger multilingual capabilities, not only is code-switching also present, but the response language is also diverse, possibly Chinese, English, or the language of the question**. Therefore, in summary, the statement should be revised to *"Notably, under the unconstrained response-language setting, models often outperform the English-only baseline. We observe that this is enabled by a flexible reasoning space, manifested in the use of multiple response languages and code-switching—where responses mix surface entities (e.g., names of people or places) from question with English or Chinese."* This has been corrected in the new version.
>
> When evaluating mGRPO, we also used the unconstrained response-language setting and observed that regardless of the language of the question, **mGRPO tends to respond in English. To understand the reasons for this phenomenon, we analyzed the changes in response language and entropy during training.** We found that the response language gradually decreased from high linguistic consistency (generating rich multilingual thinking) during training, until by the 5th epoch, almost all responses were in English. This change resulted in the model having high entropy initially, allowing it to learn a richer policy space, and then gradually decreasing entropy to stabilize the model's capabilities in a confident language space (English). Therefore, even without language restrictions, mGRPO still tends to respond in English. See Appendix H of the new version for a detailed analysis.
>
> **Based on your suggestion, we further observed whether the code-switching phenomenon still exists under the unconstrained setting.**
>
> Taking responses in Thai as an example:
>
> ● **Question in Thai** : ฟินน์ดูลูกอ๊อด 11 ตัวว่ายน้ำอยู่ในบึง ทันใดนั้นเขาก็เห็นลูกอ๊อด 6 ตัวในกลุ่มออกมาจากที่ซ่อนใต้ใบบัว ต่อมาเขาเห็นลูกอ๊อด 2 ตัวในกลุ่มเข้าไปหลบอยู่ใต้ก้อนหิน ตอนนี้ฟินน์จะเห็นลูกอ๊อดในบึงกี่ตัว
> (Translation: Finn watches 11 tadpoles swimming in a swamp. Suddenly, he sees 6 tadpoles in the group come out from their hiding place under a lotus leaf. Later, he sees 2 tadpoles in the group hiding under a rock. How many tadpoles will Finn see in the swamp now?)
>
> ● **Groud-Truth**: 15
>
> ● **Answer from Qwen2.5-7B-Instruct**: ตามลำดับขั้นตอน:  (In order of steps:)
> 1. ฟินน์เห็นลูกอ๊อด 11 ตัวอยู่ในบึง (1. Finn sees 11 tadpoles in the swamp.)
> 2. เขาเห็นลูกอ๊อด 6 ตัวในกลุ่มออกมาจากที่ซ่อน (หมายความว่า 11 - 6 = 5 ตัวยังคงอยู่ในบึง)
> (2. He sees 6 tadpoles in the group come out from their hiding place (meaning 11 - 6 = 5 still in the swamp).)
> 3. ต่อมาเขาเห็นลูกอ๊อด 2 ตัวในกลุ่มเข้าไปหลบอยู่ใต้ก้อนหิน (หมายความว่า 5 - 2 = 3 ตัวยังคงอยู่ในบึง)
> (3. Later, he sees 2 tadpoles in the group hiding under a rock (meaning 5 - 2 = 3 still in the swamp).)
> ดังนั้น, ตอนนี้ฟินน์จะเห็นลูกอ๊อดในบึง 3 ตัว
> (Therefore, Finn will now see 3 tadpoles in the swamp.)
> \#\#\#\# 3
> ● **Answer in mGRPO**: Let's break this down step by step:
> 1. Initially, Finn saw 11 ลูกอ๊อด (tadpoles) swimming in the pond.
> 2. Then, 6 ลูกอ๊อด (tadpoles) emerged from hiding under the lotus leaves. So the visible count becomes:
> 11 + 6 = 17 tadpoles.
> 3. Afterwards, 2 ลูกอ๊อด (tadpoles) hid under a stone. Subtracting them:
>    17 − 2 = 15 tadpoles.<br><br>So, the final number of ลูกอ๊อด Finn can see is 15.
> \#\#\#\# 15
>
> It can be seen that some code-switching still occurs in mGRPO. **This indicates that even though mGRPO primarily uses English responses, it still utilizes its "flexible thinking space" at test time due to its multilingual thinking training.**
>
> Thank you for suggesting that we conduct a special analysis of the unconstrained setting, which helped us identify inappropriate expressions and conduct a more in-depth analysis.

---

> > ### Author Response · Authors · 2025-11-22
> > **Response to Reviewer 8cad (2/2)**
> >
> > ### **R2: Response to "granular reward design" and related Q2**
> >
> > I understand your concern; a simple reward design can easily lead the model to seek shortcuts, i.e., directly outputting the answer or giving an answer after random thinking. We have actually considered this issue, but we missed a relevant detailed description: "To prevent the model from taking shortcuts, e.g., generating minimal text before directly outputting the answer, we additionally enforce a minimum length constraint of 100 characters for the reasoning content within the *[Thinking]* section as part of the format reward." This has been added to Lines 222-225 of the new version.**By limiting the length of the output characters in [Thinking], we prevent the model from directly giving an answer without careful consideration.**
> >
> > We incorporated your suggestions:  (1) assign a partial reward (0.5) for responses with reasoning but no answer;   (2) reduce the format reward (FR) to [0.8, 0.5, 0.1]—since both format and correctness rewards were initially set to 1, the model might prioritize format over answer accuracy. The results on Qwen2.5-1.5B-Instruct shown in below:
> >
> > | MGSM | AVG | HRL | URL | EN | DE | FR | ES | RU | ZH | JA | TH | TE | BN | SW |
> > |--|--|--|--|--|--|--|--|--|--|--|--|--|--|--|
> > | Qwen2.5-1.5B-Instruct| 26.1 | 31.5 | 14.1 | 41.2 | 24.4 | 29.6 | 34.4 | 35.2 | 40.0 | 25.6 | 29.2 | 6.4 | 17.6 | 3.2 |
> > | mGRPO | **51.0** | **62.5** | **26.9** | 78.4 | **61.6** | **65.6** | **66.8** | **66.0** | **65.6** | **49.6** | **53.6** | **12.8** | **34.0** | **7.2** |
> > | mGRPO w/ (1) | 47.9 | 59.5 | 22.4 | **80.0** | 61.2 | 63.2 | 63.6 | 61.6 | 62.4 | 44.8 | 49.2 | 7.2 | 28.8 | 4.4 |
> > | mGRPO w/ (2)FR=0.8 | 49.1 | 60.3 | 25.5 | 76.0 | 61.2 | 64.0 | 65.2 | 63.2 | 62.4 | 46.0 | 50.8 | 10.0 | **34.4** | 6.8 |
> > | mGRPO w/ (2)FR=0.5 | 48.4 | 60.7 | 23.9 | 73.2 | 60.4 | 62.0 | 63.2 | 65.2 | **65.6** | 47.6 | 48.8 | 10.8 | 30.0 | 6.0 |
> > | mGRPO w/ (2)FR=0.1 | 47.6 | 59.6 | 22.8 | 75.2 | 58.8 | 60.8 | 62.0 | 63.6 | 63.6 | 48.8 | 47.6 | 8.0 | 30.8 | 4.8 |
> >
> > Experimental results show that **mGRPO achieves optimal performance when trained with the standard format `[thinking] + [Answer]`, where the `[thinking]` segment contains at least 100 characters and a fixed reward value of 1 is used.**
> >
> > Regarding the concern about incorrect reasoning leading to correct answers, we note that this could be mitigated via an *LLM-as-a-judge* pipeline for finer-grained process evaluation. In our setting—using moderately simple math problems—such cases were rarely observed. While this may become more relevant with harder tasks, it falls outside the core scope of our work, and thus we did not pursue further experiments. We sincerely thank you for this insightful comment.
> >
> > ### **R3: Response to “Is the multilingual-thinking-as-scaffolding a temporary effect?”**
> >
> > Thank you very much for your valuable suggestion. We extended the training of mGRPO and GRPO from epoch=5 to epoch=10 to observe performance changes and analyzed whether the impact of multilingual thinking is temporary. **The results shown in Figure 8 of new version indicats that mGRPO consistently maintained its performance advantage after 5 epochs**. It indicates that multilingual thinking in early training establishes a stronger foundation and effectively enhances the model’s reasoning capability. Detailed experimental analysis and conclusions can be found in the new version of Appendix H.2.
> >
> > ### **R4: Response to “Could the authors comment on how much "base" multilingualism is necessary for mGRPO to be effective?”**
> >
> > Given the differing baseline capabilities of Llama and Qwen, we evaluated our method separately on both. Notably, even on Llama—whose multilingual ability is weaker than Qwen’s—our approach still yields a clear improvement (+3.53% over GRPO on MGSM), indicating that strong inherent multilingual capacity in the base model is not required. This is further supported by ablation studies in Section 5.1 using Qwen2.5-Instruct variants of varying sizes, which confirm that **model-scale differences in multilingual proficiency do not undermine the effectiveness of our method**.
> >
> > ### **R5: Response to “performance drops with more rollouts (e.g., n=8 or n=10)”**
> >
> > Our language set includes both high- and low-resource languages. Larger rollouts increase the likelihood of sampling low-resource responses (e.g., Swahili, Telugu), which are inherently weaker in the model. When such responses are rare, they can act as *helpful noise*—highlighting stronger languages and enabling cross-lingual transfer. However, as their proportion grows, even small per-sample advantages accumulate, turning this noise *harmful*. Therefore, increasing the rollout n will bring more noise than helpful signals from some low-quality non-English responses.
> >
> > ---
> >
> > I hope my reply has answered your questions. If there is anything unclear or omitted, I look forward to your further replies.

---

### Official Review · Reviewer_Bkj8 · 2025-11-01

**Soundness:** 3
**Presentation:** 3
**Contribution:** 2
**Rating:** 4
**Confidence:** 5

**Summary:**

This paper proposes Multilingual Group Relative Policy Optimization (mGRPO), a reinforcement learning framework, designed to enhance reasoning capabilities of LLMs through multilingual thinking. Motivated by a preliminary experiment showing that multilingual reasoning can outperform monolingual (English-only) reasoning, the authors extend the standard GRPO framework by introducing a multilingual rollout strategy. During training, the model generates a set of multilingual reasoning trajectories under both language-constrained and language-unconstrained prompts, and optimizes them via GRPO. Experiments on four reasoning benchmarks—MGSM, MATH500, PolyMath, and X-CSQA, demonstrate consistent improvements over the baseline GRPO and other variants.

**Strengths:**

1. **Clear Insight with Preliminary Validation**. This paper has conceptual clarity and a well-motivated insight derived from the “Polyglot Thinking Experiment.” The preliminary analysis convincingly shows that (1) reasoning in certain non-English languages can outperform English; and (2) granting the model the freedom to choose its reasoning language leads to measurable gains. These findings provide direct empirical evidence for the motivation of leveraging multilingual thinking, making the subsequent methodological development logical.

2. **In-depth Experimental Analysis**. Beyond reporting end-task performance, the paper includes a series of insightful analyses to interpret model behavior. The ablation studies highlight the complementary role of language-constrained and unconstrained reasoning paths, underscoring the necessity of both. The logit-lens analysis reveals that the mGRPO-trained model tends to integrate multilingual reasoning signals into an English-centric latent space, enabling it to perform high-quality reasoning with fewer non-English tokens. This offers an interpretable explanation for why the proposed method works.

**Weaknesses:**

1. **Methodology is Rather Incremental** While the framework is conceptually sound, its algorithmic novelty is limited. The core contribution lies in a prompt-based multilingual data generation module on top of GRPO, while the optimization pipeline itself remains unchanged, still relying on the standard GRPO update and basic reward formulation. The work does not deeply explore richer forms of multilingual signals, such as logical consistency across languages or language-specific reasoning properties. As a result, the method feels more like a creative application of existing techniques rather than a new learning framework.

2. **Baselines Could Be Expanded** Although the paper compares against several reasonable baselines, it omits some relevant recent works that extend GRPO like DAPO[1] or GSPO[2]. Including these would make the empirical evaluation more comprehensive and the claims more robust.

3. **Eventual Reliance on English Reasoning** The analysis section notes that the trained model ultimately converges toward English reasoning during inference. The authors interpret this as knowledge fusion into an English-dominant latent space. However, this observation also reveals a limitation: the framework mainly leverages multilingual reasoning to strengthen English reasoning rather than fostering truly language-agnostic reasoning abilities. This weakens the central hypothesis that multilingual thinking inherently enhances reasoning. Moreover, it implies that the model may still struggle to reason effectively in specific non-English languages. While the appendix (mGRPO-lang) explores a mitigation strategy, the admitted trade-off in performance suggests that practical multilingual applicability remains limited.

4. **Lack of Stability and Statistical Validation** Key experimental results (e.g., Table 1) appear to be reported from single runs, without standard deviation, confidence intervals, or statistical significance analysis. This omission raises concerns about the robustness of the reported improvements. Multiple independent runs with averaged scores and error bars would significantly strengthen the reliability and reproducibility of the results.

[1] Dapo: An open-source llm reinforcement learning system at scale.

[2] Group sequence policy optimization.

**Questions:**

See Weaknesses.

---

> ### Author Response · Authors · 2025-11-22
> **Response to Reviewer Bkj8**
>
> We sincerely appreciate your acknowledgment of the **sound motivation** and **thorough analysis** in our work. Below, we address each of your comments and concerns in turn.
>
> ---
>
> ### **R1: Response to “Methodology is incremental”**
> We sincerely appreciate the reviewer’s concern regarding algorithmic novelty.
>
>  The core innovation of our method stems from the findings of the *Polyglot Thinking Experiment*, which naturally motivate a reinforcement learning framework that treats *multilingual thinking* as an action space to improve model reasoning capabilities. This perspective aligns with your acknowledgment in Strength 1. Furthermore, mGRPO achieves an average improvement of **+7.5%** across multiple mathematical and commonsense reasoning benchmarks. **These results substantiate that our framework, while simple in design, yields meaningful and consistent performance gains.** We maintain that methodological innovation can be demonstrated through conceptual clarity and empirical impact, not solely through architectural complexity.
>
>  We agree that exploring richer forms of multilingual signals, as you suggested, represents a valuable direction for future research. Accordingly, we have added this point in Lines 1083 to 1085 to explicitly indicate it as a promising extension.
>
> ### **R2: Response to “Expend Baselines”**
> The DAPO and GSPO methods you mentioned are variants that improve upon the original GRPO framework. **Our approach is orthogonal and general: it can be applied not only to GRPO but also to its extensions, including DAPO and GSPO.** To support this claim, we have additionally conducted experiments with GSPO and its multilingual-thinking-enhanced counterpart, mGSPO. The results are shown in the table below:
>
> | MGSM | AVG | HRL | URL |
> |--|--|--|--|
> | *Qwen2.5-7B-Instruct* | | | |
> | Base | 67.2 | 75.7 | 48.8 |
> | xRFT | 68.5 | 81.1 | 43.2 |
> | LIDR | 69.6 | 79.1 | 50.3 |
> | MAPO | 66.3 | 75.8 | 47.4 |
> | GRPO | 73.0 | 81.1 | 56.6 |
> | **mGRPO** | **75.9** | **84.4** | **58.7** |
> | | | | |
> | GSPO | 67.7 | 76.8 | 49.5 |
> | **mGSPO** | **74.3** | **83.9** | **55.5** |
>
>  Results on the MGSM benchmark indicate that GSPO underperforms GRPO-based methods, likely because its sequence-level credit assignment is suboptimal for multi-step reasoning tasks. Nevertheless, mGSPO achieves an average accuracy improvement of +6.6% over GSPO and +1.3% over GRPO. This confirms that **multilingual thinking consistently enhances reasoning capability across different reinforcement learning optimization granularities**.
>
> ### **R3: Response to “Eventual Reliance on English Reasoning”**
> Regarding the observation that the final reasoning model tends to favor English for internal reasoning, we briefly described a simple test-time strategy in the submitted version (Lines 466–468 and Appendix H): appending a language-specific prefix—such as *“Okay,”* for English or *“D’accord,”* for French—to the input question to steer mGRPO toward language-aligned reasoning. This approach enables multilingual reasoning with only a marginal performance drop, as reported in Table 14 of the original submission. In the revised manuscript, these results are presented in Table 14 of Appendix I.
>
> | Model | AVG | HRL | URL |
> |--|--|--|--|
> | **Accuracy** | | | |
> | Qwen2.5-7B-Instruct | 67.2 | 75.7 | 48.8 |
> | ------w/ prefix | 66.8 | 76.5 | 46.4 |
> | GRPO | 73.0 | 81.1 | 56.6 |
> |  ------w/ prefix | 72.4 | 82.4 | 52.7 |
> | mGRPO | **75.9** | **84.4** | **58.7** |
> | ------w/ prefix | 74.3 | 83.7 | 55.7 |
> | | | | |
> | **Language Consistency** | | | |
> | Qwen2.5-7B-Instruct | 68.4 | 92.6 | 24.3 |
> |  ------w/ prefix | 99.7 | 99.9 | 99.6 |
> | GRPO | 17.8 | 14.4 | 2.3 |
> |  ------w/ prefix | 99.7 | 99.9 | 99.4 |
> | mGRPO | 9.6 | 0.2 | 1.1 |
> |  ------w/ prefix | 99.1 | 100.0 | 97.4 |
>
> It show that, although mGRPO exhibits a preference for generating reasoning steps in English, **high-quality multilingual reasoning can still be achieved through straightforward test-time prompting strategies, without complex architectural modifications.**
>
> About mGRPO_lang, it serves as an exploratory implementation of the intuitive idea of incorporating a *language consistency reward*. However, designing an effective and stable language consistency reward requires careful tuning. While this direction merits further investigation (as noted in Line 1130 -1150 of the revised manuscript Appendix I), it lies beyond the primary scope of the present work, which focuses on utilzing the *multilingual thinking* as a general and lightweight enhancement to RL-based reasoning.
>
> **And we analysis the *multilingual thinking* behavior during training in Appendix H of new version.**
>
> ### **R4: Response to “Stability and Statistical Validation”**
> As stated in Line 309 (coresponding to Line 315 of revised version), our evalution is conducted under **greedy decoding**, ensuring that the outputs are deterministic and fully reproducible.
>
> ---
>
> We look forward to your further comments.

---

### Author Response · Authors · 2025-11-22
**Summary of review comments and revision notes for the new version**

We are very grateful to all the reviewers for their careful review of our work and for recognizing it in the following aspects:
- **Reviewer Bkj8 and Reviewer 8cad acknowledge that the paper is grounded in clear empirical motivation** from the *Polyglot Thinking Experiment*, demonstrating that language-unconstrained reasoning or reasoning in certain non-English languages, may outperform English reasoning.

- **Reviewer 8cad, Reviewer BddF, and Reviewer EMqQ further highlight that mGRPO effectively operationalizes this insight** by combining unconstrained and language-constrained multilingual thinking rollouts, thereby expanding the model’s reasoning space and achieving SOTA results with a lightweight training setup.

- **Reviewer Bkj8 notes that the paper also provides in-depth experimental analysis** to explain *why* the proposed method works.

---

We sincerely thank the reviewers for their constructive feedback, which significantly improved the paper. In the revised submission, we have addressed all comments—not only correcting typos, but also implementing the following substantive revisions:

  1. **Clarified related description and added analysis with unconstrained response-language setting** (Lines 91–95; Appendix H.1, Lines 994–1019):

     The analysis indicated that:

- mGRPO under unconstrained response language setting, the response gradually shift from multilingual to English-dominated reasoning during training process, supporting our claim that *multilingual thinking* is internalized and distilled into English reasoning.
 - Though outputs are mostly in English, code-switching (e.g., retention of non-English entities from the question) is observed, evidence of thinking from flexible multilingual reasoning space.

  2. **Supplementing details of reward design** (Lines 222–224):
     To prevent shortcut behaviors (e.g., less text generation before answer), we require reasoning traces to exceed 100 characters to qualify for format reward.

  3. **Extended analysis of *multilingual thinking* in training** (Appendix H):
     (i) Language consistency declines over epochs, converging to English by epoch 5 and remaining stable thereafter;
     (ii) Despite convergence to English, mGRPO consistently outperforms GRPO beyond epoch 5—demonstrating lasting impact of multilingual thinking;
     (iii) Entropy analysis shows higher initial entropy in mGRPO, enabling broader exploration, followed by stable convergence to a confident policy.

  4. **Added GSPO and multilingual GSPO (mGSPO) experiments** (Appendix K):
     mGSPO outperforms GSPO, confirming transferability of our approach across RL frameworks.

  5. **Expanded future work** (Appendices H & I):
     We outline promising directions, e.g., refined language-consistency rewards or designed cross-lingual rewards, interpretability studies on language-reasoning interplay, to encourage community exploration. Our work establishes a simple yet effective proof-of-concept; deeper investigation remains valuable.

In the newly submitted manuscript, we have marked the revised text or newly added sections in blue.

---

### Meta-Review · Area_Chair_kVjT · 2026-01-04

**Summary:**

Overall, this paper has undergone review by four experts in the field, who assigned scores of 4, 6, 2, and 4, respectively.

The AC has checked the rebuttal and found that several concerns are still not well addressed. For instance, while the greedy-based evaluation may produce stable results, it does not necessarily reflect the model’s true performance; results averaged over multiple runs would be significantly more convincing, following the common practice in other reasoning works. Moreover, Reviewer BddF highlighted a critical limitation: the method’s inability to control the language of thought without compromising reasoning quality, and he/she may not change the score. The AC also agrees with the reviewers’ assessment that the contribution is incremental, primarily extending the existing GRPO pipeline through prompt-based data generation, and appears to be a specific application of GRPO to multilingual settings via prompt engineering. Therefore, the AC believes that the response does not sufficiently overturn the generally negative evaluation.

In light of the overall scores and the unresolved issues raised during review, and ICLR’s highly competitive acceptance rate this year, the AC regrets to recommend rejection.

**Reviewer Concerns:**

Concerns Addressed:

1. Baseline compared: The authors in the rebuttal have provided a comparison of more baselines to make the paper results more robust.

2. Eventual Reliance on English Reasoning: The authors in the rebuttal have addressed this issue by providing the results that high-quality multilingual reasoning can be achieved.

3. Analysis/Typos and Implementation details: The authors have provided the details in the rebuttal.

Remaining Concerns:

1. Novelty: The AC agrees with the reviewer that the novelty is limited in this paper, as the prompt-based multilingual data generation is somewhat a trivial improvement on GRPO. The authors' rebuttal does not convince the AC, and the AC believes that the reviewer may not be willing to change the score.

2. Statistical evaluation and performance drop yielded by multiple rollouts: The authors have clarified that the experiments were done with greedy decoding. However, the AC believes that the results obtained by averaging the results from multiple runs would be more convincing, as is adopted in most other reasoning works.

3. The performance drop caused by the consistency reward: the reviewer found that, for a paper targeted on multilingual capabilities, the inability to control the language of thought without degrading reasoning is a primary limitation. This is still a major issue for voting to reject this paper.

**Reviewer Scores:**

The scores are 4/6/2/4.

Reviewer Bkj8 (4): The score would likely remain unchanged, or at most increase very slightly, but not enough to clearly shift their overall recommendation, and the novelty/contribution of this paper is incremental to the GRPO pipeline.

Reviewer 8cad (6): This reviewer was generally positive, but would not mind rejection. The AC believes that the score would largely remain the same after rebuttal.

Reviewer BddF (2): This reviewer replied that his/her concern was further confirmed during the rebuttal, and the AC agrees with this point. The reviewer has stated that the score will remain.

Reviewer EMqQ (4): This reviewer shared concerns similar to those raised by Reviewer BddF. While some clarifications were provided in the rebuttal, the AC does not expect a increse on the score.

---

### Decision · Program_Chairs · 2026-01-26

Reject